# Noncoding mutations target *cis*-regulatory elements of the *FOXA1* plexus in prostate cancer

Stanley Zhou [1,2], James R. Hawley [1,2], Fraser Soares [1], Giacomo Grillo[1], Mona Teng[1,2], Seyed Ali Madani Tonekaboni[1,2], Junjie Tony Hua[1,2], Ken J. Kron[1], Parisa Mazrooei[1,2], Musaddeque Ahmed[1], Christopher Arlidge[1], Hwa Young Yun[1], Julie Livingstone [3], Vincent Huang [3], Takafumi N. Yamaguchi [3], Shadrielle M.G. Espiritu[3], Yanyun Zhu[4], Tesa M. Severson[4], Alex Murison[1], Sarina Cameron[1], Wilbert Zwart[4,5], Theodorus van der Kwast [6], Trevor J. Pugh [1,2,3], Michael Fraser[3], Paul C. Boutros [2,3,7,8,9,10,11], Robert G. Bristow [1,12,13,14,15], Housheng Hansen He[1,2] & Mathieu Lupien [1,2,3]*

Prostate cancer is the second most commonly diagnosed malignancy among men worldwide. Recurrently mutated in primary and metastatic prostate tumors, *FOXA1* encodes a pioneer transcription factor involved in disease onset and progression through both androgen receptor-dependent and androgen receptor-independent mechanisms. Despite its oncogenic properties however, the regulation of *FOXA1* expression remains unknown. Here, we identify a set of six *cis*-regulatory elements in the *FOXA1* regulatory plexus harboring somatic single-nucleotide variants in primary prostate tumors. We find that deletion and repression of these *cis*-regulatory elements significantly decreases *FOXA1* expression and prostate cancer cell growth. Six of the ten single-nucleotide variants mapping to *FOXA1* regulatory plexus significantly alter the transactivation potential of *cis*-regulatory elements by modulating the binding of transcription factors. Collectively, our results identify *cis*-regulatory elements within the *FOXA1* plexus mutated in primary prostate tumors as potential targets for therapeutic intervention.

[1] Princess Margaret Cancer Centre, University Health Network, Toronto, ON, Canada. [2] Department of Medical Biophysics, University of Toronto, Toronto, ON, Canada. [3] Ontario Institute for Cancer Research, Toronto, ON, Canada. [4] Division of Oncogenomics, Oncode Institute, the Netherlands Cancer Institute, Amsterdam, The Netherlands. [5] Laboratory of Chemical Biology and Institute for Complex Molecular Systems, Department of Biomedical Engineering, Eindhoven University of Technology, Eindhoven, The Netherlands. [6] Department of Laboratory Medicine and Pathobiology, University of Toronto, Toronto, ON, Canada. [7] Department of Pharmacology and Toxicology, University of Toronto, Toronto, CA, Canada. [8] Department of Human Genetics, University of California, Los Angeles, CA, USA. [9] Department of Urology, University of California, Los Angeles, CA, USA. [10] Institute for Precision Health, University of California, Los Angeles, CA, USA. [11] Jonsson Comprehensive Cancer Center, University of California, Los Angeles, CA, USA. [12] Department of Radiation Oncology, University of Toronto, Toronto, ON, Canada. [13] CRUK Manchester Institute and Manchester Cancer Research Centre, Manchester, UK. [14] Division of Cancer Sciences, Faculty of Biology, Health and Medicine, University of Manchester, Manchester, UK. [15] The Christie NHS Foundation Trust, Manchester, UK. *email: mlupien@uhnresearch.ca

Prostate cancer is the second most commonly diagnosed cancer among men with an estimated 1.3 million new cases worldwide in 2018[1]. Although most men diagnosed with primary prostate cancer are treated with curative intent through surgery or radiation therapy, treatments fail in 30% of patients within 10 years[2] resulting in a metastatic disease[3]. Patients with metastatic disease are typically treated with anti-androgen therapies, the staple of aggressive prostate cancer treatment[4]. Despite the efficacy of these therapies, recurrence ultimately develops into lethal metastatic castration resistant prostate cancer (mCRPC)[4]. As such, there remains a need to improve our biological understanding of prostate cancer development and find novel strategies to treat patients.

Sequencing efforts identified coding somatic single-nucleotide variants (SNVs) mapping to FOXA1 in up to 9%[5–10] and 13%[9–11] of primary and mCRPC patients, respectively. These coding somatic SNVs target the Forkhead and transactivation domains of FOXA1[12], altering its pioneering functions to promote prostate cancer development[10,13]. Outside of coding SNVs, whole-genome sequencing also identified somatic SNVs and indels in the 3'UTR and C-terminus of FOXA1 in ~12% of mCPRC patients[14]. In addition to SNVs, the FOXA1 locus is a target of structural rearrangements in both primary and metastatic prostate cancer tumors, inclusive of duplications, amplifications, and translocations[9,10]. Taken together, FOXA1 is recurrently mutated taking into account both its coding and flanking noncoding sequences across various stages of prostate cancer development.

FOXA1 serves as a pioneer transcription factor (TF) that can bind to heterochromatin, promoting its remodeling to increase accessibility for the recruitment of other TFs[15]. FOXA1 binds to chromatin at cell-type specific genomic coordinates facilitated by the presence of mono- and dimethylated lysine 4 of histone H3 (H3K4me1 and H3K4me2) histone modifications[16,17]. In prostate cancer, FOXA1 is known to pioneer and reprogram the binding of the androgen receptor (AR) alongside HOXB13[18]. Independent from its role in AR signaling, FOXA1 also regulates the expression of genes involved in cell cycle regulation in prostate cancer[19–21]. For instance, FOXA1 co-localizes with CREB1 to regulate the transcription of genes involved in cell cycle processes, nuclear division, and mitosis in mCRPC[19–25]. FOXA1 has also been shown to promote feed-forward mechanisms to drive disease progression[26,27]. Hence, FOXA1 contributes to AR-dependent and AR-independent processes favouring prostate cancer development.

Despite the oncogenic roles of FOXA1, therapeutic avenues to inhibit its activity in prostate cancer are lacking. In the breast cancer setting for instance, the use of cyclin-dependent kinases inhibitors have been suggested based on their ability to block FOXA1 activity on chromatin[28]. As such, understanding the governance of FOXA1 mRNA expression offers an alternative strategy to find modulators of its activity. Gene expression relies on the interplay between distal cis-regulatory elements (CREs), such as enhancers and anchors of chromatin interaction, and their target gene promoter(s)[29]. These elements can lie tens to hundreds of kilobases (kbp) away from each other on the linear genome but physically engage in close proximity with each other in the three-dimensional space[30]. By measuring contact frequencies between loci through the use of chromatin conformation capture-based technologies, it enables the identification of regulatory plexuses corresponding to sets of CREs in contact with each other[31,32]. By leveraging these technologies, we can begin to understand the three-dimensional organization of the prostate cancer genome and delineate the FOXA1 regulatory plexus.

Here, we integrate epigenetics and genetics from prostate cancer patients and model systems to delineate CREs establishing the regulatory plexus of FOXA1. We functionally validate a set of six mutated CREs that regulate FOXA1 mRNA expression. We further show that SNVs mapping to these CREs are capable of altering their transactivation potential, likely through modulating the binding of key prostate cancer TFs.

## Results

**FOXA1 is essential for prostate cancer proliferation.** We interrogated FOXA1 expression levels across cancer types. We find that FOXA1 mRNA is consistently the most abundant in prostate tumors compared with 25 other cancer types across patients (Fig. 1a), ranking in the 95th percentile for 492 of 497 prostate tumors profiled in TCGA (Supplementary Fig. 1a). Using the same data set we also find that FOXA1 is the most highly expressed out of 41 other Forkhead Box (FOX) factors in prostate tumors (Supplementary Fig. 1b). We next analyzed expression data from DEPMAP and observed FOXA1 to be most highly expressed in prostate cancer cell lines compared with cell lines of other cancer types (Supplementary Fig. 2a). Among the eight prostate cancer cell lines in the dataset (22Rv1, DU145, LNCaP, MDA-PCa-2B, NCI-H660, PrECLH, PC3, and VCaP), FOXA1 mRNA abundance is above the 90th percentile in all but one cell line (PrECLH) compared with the >56,000 protein coding and non-protein coding genes profiled (Supplementary Fig. 2b). These new results gained from the TCGA and DEPMAP validate previous understanding that FOXA1 is one of the highest expressed genes in prostate cancer[33].

Following up on FOXA1 mRNA expression levels, we interrogated the essentiality of FOXA1 for prostate cancer cell growth. RNAi-mediated essentiality screens compiled in DEPMAP show that FOXA1 lies in the 94th percentile across six of the eight available prostate cancer cell lines: 22Rv1, LNCaP, MDA PCa 2B, NCI-H660, PC3, and VCaP cells (Fig. 1b, c). The median RNAi-mediated essentiality score for all prostate cell lines is significantly lower than all other cell lines, suggesting that FOXA1 is especially essential for prostate cancer cell proliferation (permutation test, $p = 1 \times 10^{-6}$, see Methods) (Supplementary Fig. 3a). Growth assays in LNCaP and VCaP cells following FOXA1 knockdown using two independent siRNAs (Fig. 1d, Supplementary Fig. 3b) show significant growth inhibition in LNCaP (siRNA #1: fourfold, siRNA #2: 3.35-fold) and VCaP (siRNA #1: 8.7-fold, siRNA #2: twofold) cells 5 days post transfection (Mann–Whitney U test, $p < 0.05$; Fig. 1e, f). In accordance with previous reports, our results using essentiality datasets followed by knockdown validation reveals that FOXA1 is oncogenic and essential for prostate cancer cell proliferation.

**Identifying putative FOXA1 CREs.** The interweaving of distal CREs with target gene promoters establishes regulatory plexuses with some to be ascribed to specific genes[31,32]. Regulatory plexuses stem from chromatin interactions orchestrated by various factors including ZNF143, YY1, CTCF, and the cohesin complex[34–36]. Motivated by the oncogenic role of FOXA1 in prostate cancer, we investigated its regulatory plexus controlling its expression. According to chromatin contact frequency maps generated from Hi-C assays performed in LNCaP prostate cancer cells, FOXA1 lies in a 440 kbp TAD (chr14: 37720002–38160000 ± 40 kbp adjusting for resolution) (Fig. 2a). By overlaying DNase-seq data from LNCaP prostate cancer cells, there are a total of 123 putative CREs reported as DNase I hypersensitive sites (DHS) that populate this TAD (Fig. 2a). We next inferred the regulatory plexus of FOXA1 using the cross cell-type correlation based on DNA accessibility (C3D) method[37]. C3D aggregates and draws correlation of DHS signal intensities between the cell line of choice and the DHS signal across all systems in the database[37]. Anchoring our analysis to the FOXA1 promoter and using

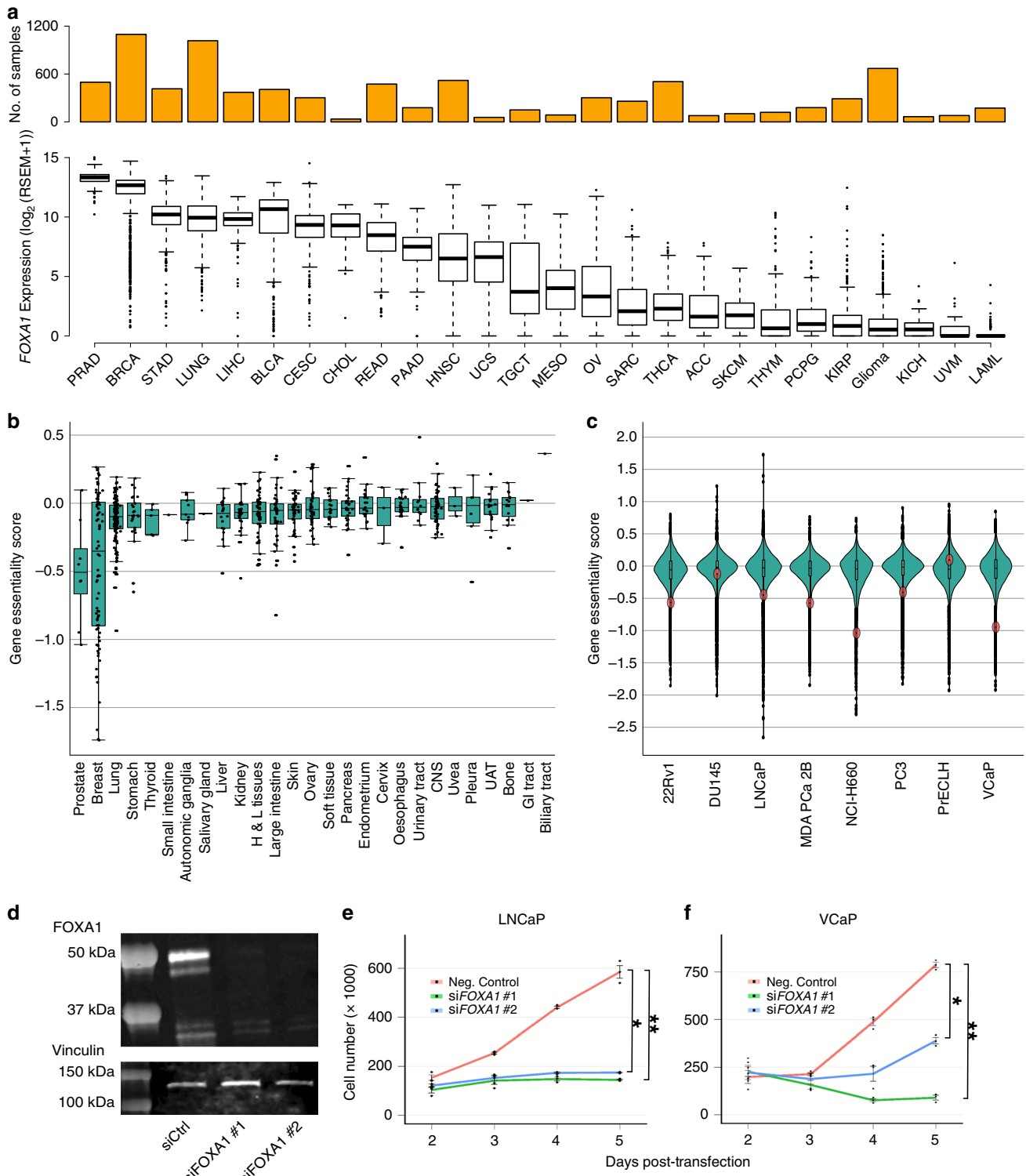

**Fig. 1 FOXA1 is highly expressed in prostate cancer and essential for prostate cancer cell proliferation. a** The mRNA expression of *FOXA1* across tumor types ($n = 26$) from RNA-seq data of TCGA. **b** *FOXA1* essentiality mediated through RNAi across various cell lines ($n = 707$) from DEPMAP. Gene essentiality scores are normalized $Z$ scores. Higher scores indicate less essential, and lower scores indicate more essential for cell proliferation. *X* axis indicate tissue of origin for each cell line tested. Each dot indicates one cell line. **c** Gene essentiality mediated through RNAi across prostate cancer cell lines ($n = 8$) from DEPMAP. Each dot indicates one gene, red indicates *FOXA1*. **d** Representative western blot against FOXA1 in LNCaP cells 5 days post transfection of non-targeting siRNA and two independent siRNA targeting *FOXA1*. **e** Cell proliferation assay conducted in LNCaP cells upon siRNA-mediated knockdown of *FOXA1* across 5 days. **f** Cell proliferation assay conducted in VCaP cells upon siRNA-mediated knockdown of *FOXA1* across 5 days. Error bars indicate ± s.d. $n = 3$ independent experiments. Mann–Whitney $U$ test, *$p < 0.05$, **$p < 0.01$.

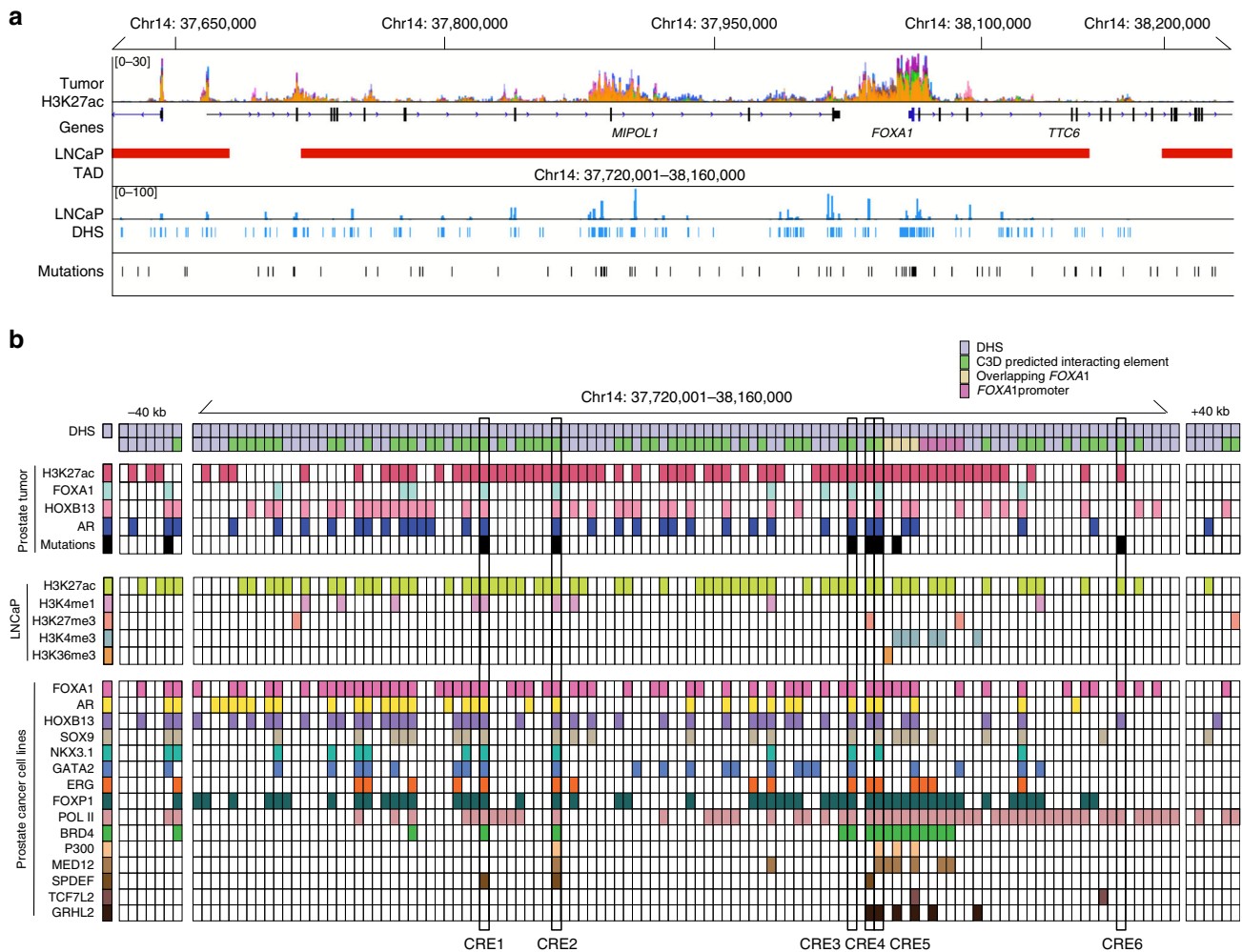

**Fig. 2 Epigenetic annotation of 14q21.1 locus and identification of *FOXA1* CREs. a** Overview of *cis*-regulatory landscape surrounding *FOXA1* on the 14q21.1 locus. H3K27ac signal track is the ChIP-seq signal overlay of 19 primary prostate tumors[38]. LNCaP Hi-C depicts the TAD structure around *FOXA1*. Mutations indicate SNVs identified in 200 primary prostate tumors[6,40]. **b** Functional annotation of putative *FOXA1* CREs using transcription factor and histone modification ChIP-seq conducted in primary tumors and prostate cancer cell lines. Annotated in the matrix are all DHS within the TAD and ± 40 kbp resolution left and right of the TAD. Putative *FOXA1* CREs targeted by noncoding SNVs for downstream validation are boxed.

accessible chromatin regions defined in LNCaP prostate cancer cells identified 55 putative CREs to the *FOXA1* regulatory plexus ($r > 0.7$) (Fig. 2b).

**Putativhe *FOXA1* CREs harbor TF-binding sites and SNVs**. To delineate the CREs that could be actively involved in the transcriptional regulation of *FOXA1*, we annotated the DHS with available ChIP-seq data for histone modifications and TFs conducted in LNCaP, 22Rv1, VCaP prostate cancer cell lines and primary prostate tumors (Fig. 2b)[18,38]. Close to 60% (33/55) of the putative *FOXA1* plexus CREs are positively marked by H3K27ac profiled in primary prostate tumors[38], indicative of active CREs in tumors (Fig. 2b)[39]. Next, considering that noncoding SNVs can target a set of CREs that converge on the same target gene in cancer[32], we overlapped the somatic SNVs called from the whole-genome sequencing across 200 primary prostate tumors to the 33 H3K27ac-marked DHS predicted to regulate *FOXA1* (Supplementary Data)[6,40]. This analysis identified 6 out of the 33 DHS marked with H3K27ac (18.2%) harboring one or more SNV(s) (10 total SNVs called from nine tumors) (Fig. 2b). We observe that these six CREs can be bound by multiple TFs in prostate cancer cells, including FOXA1, AR, and HOXB13 (Fig. 2b, Supplementary Fig. 4a–f). The Hi-C data from the

LNCaP prostate cancer cells corroborates the C3D predictions as demonstrated by the elevated contact frequency between the region harboring the *FOXA1* promoter and where the six CREs are located, compared with other loci in the same TAD (Fig. 3a). The six CREs lie in intergenic or intronic regions (Fig. 3b–h). Together, histone modifications, TF-binding sites and noncoding SNVs support that these six putative CREs are active in primary prostate cancer. The Hi-C and C3D predictions suggest that they regulate *FOXA1* expression.

**Disruption of CREs reduces *FOXA1* mRNA expression**. We next assessed the role of CREs toward *FOXA1* expression using LNCaP and 22Rv1 clones stably expressing the wild-type Cas9 protein (Fig. 4a, b). Guide RNAs (gRNAs) designed against the *FOXA1* gene (exon 1 and intron 1) served as positive control while an outside-TAD region (i.e., termed Chr14 (−)), a region on a different chromosome (the human AAVS1 safe-harbor site at the *PPP1R12C* locus[38,41]), and three regions within the TAD predicted to be excluded from the *FOXA1* plexus served as negative controls (Supplementary Data). Individual deletion of the *FOXA1* plexus CREs through transient transfection of gRNAs into the LNCaP cells (See Methods) led to significantly decreased *FOXA1* mRNA expression (ΔCRE1 ~ 29.3 ± 8.3%,

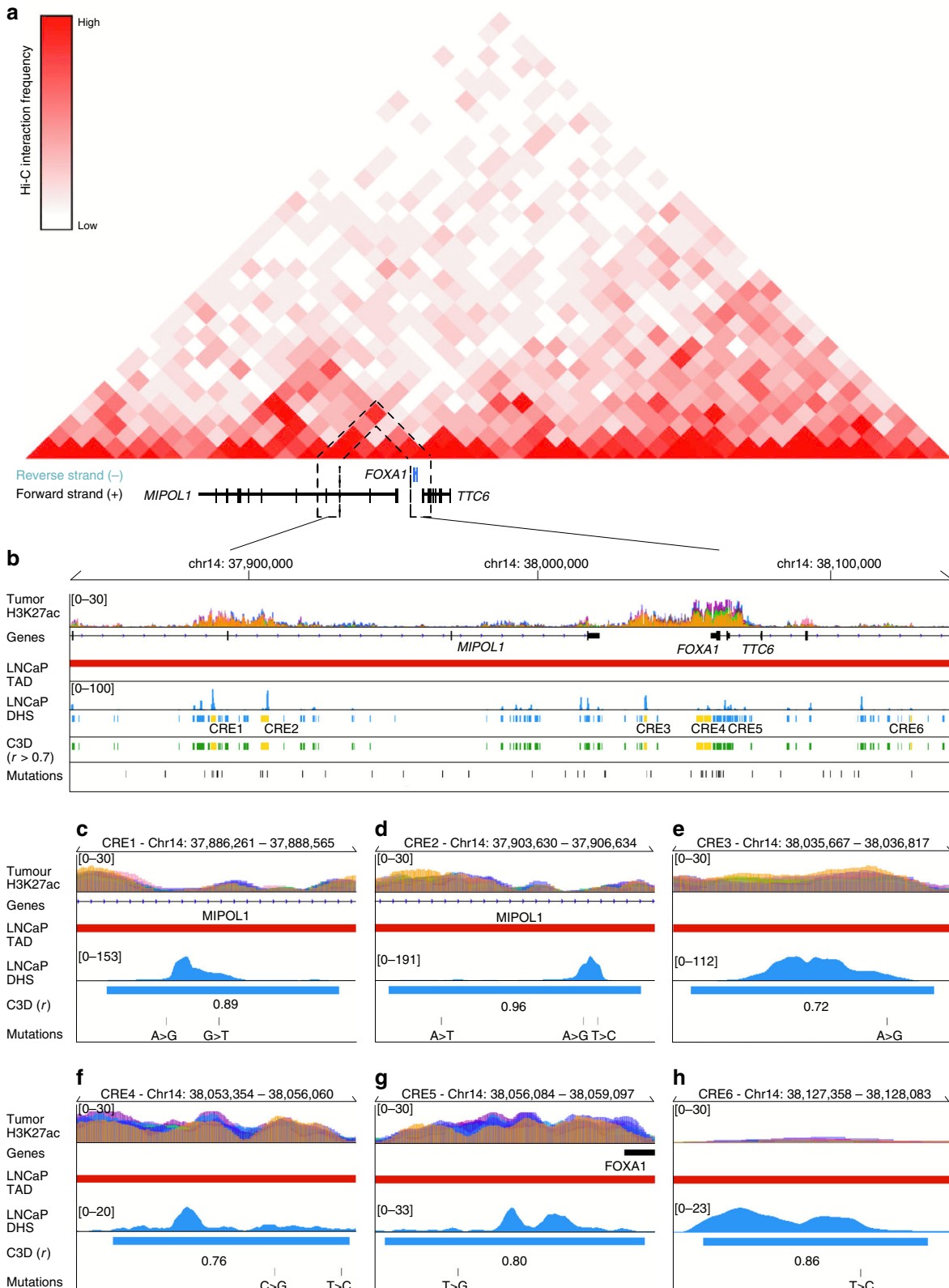

**Fig. 3 Putative CREs predicted to interact with *FOXA1* promoter. a** Hi-C conducted in LNCaP cells indicating physical interactions between putative *FOXA1* CREs and *FOXA1* promoter. Hi-C resolution is 40 kbp. **b** The six putative *FOXA1* CREs are colored in yellow. **c–h** Zoom-in of each individual putative *FOXA1* CRE. C3D[37] value is Pearson correlation of DHS signal between LNCaP and the DHS reference matrix.

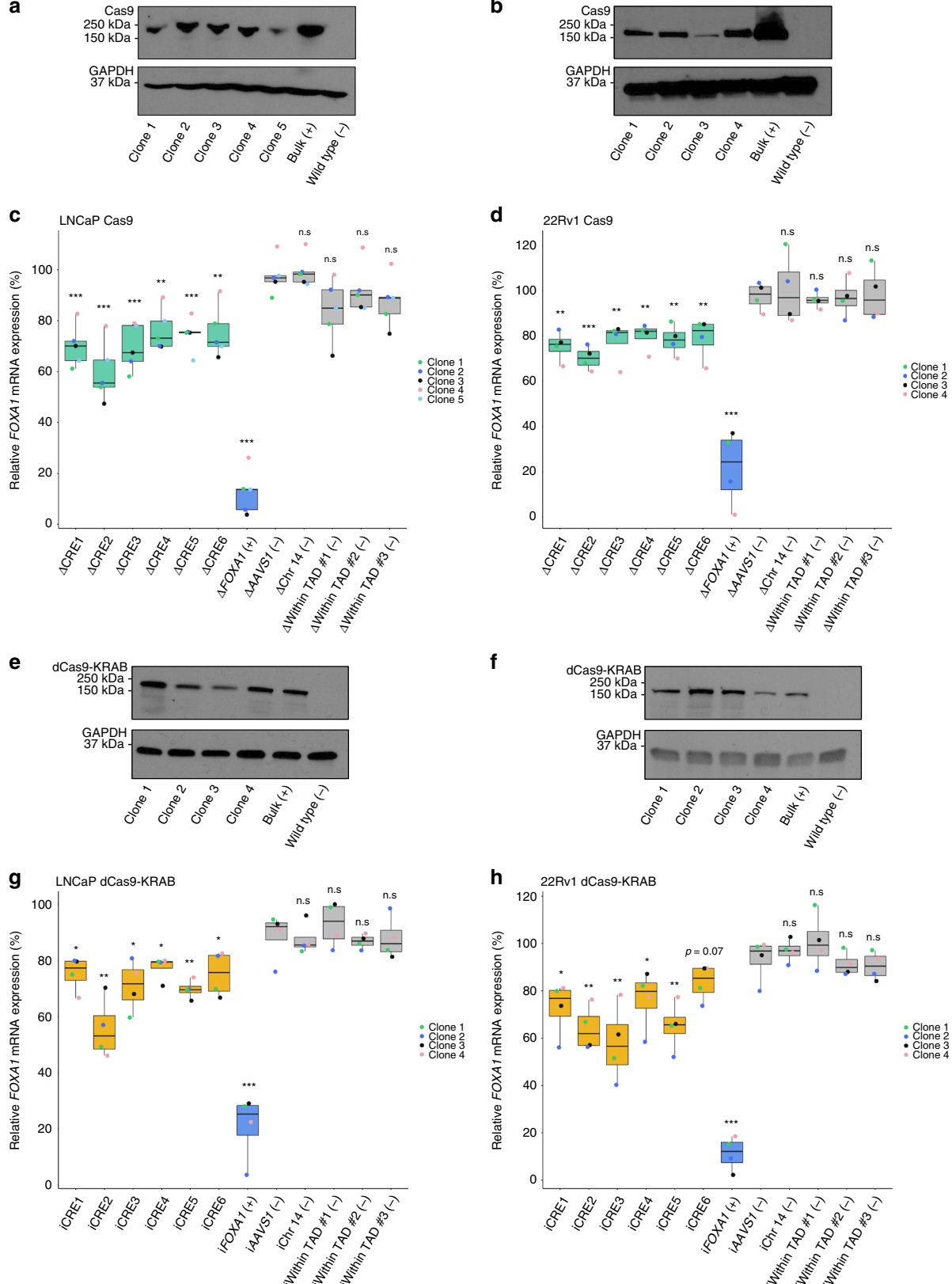

ΔCRE2 ~ 40.1 ± 11.8%, ΔCRE3 ~ 30.6 ± 9.1%, ΔCRE4 ~ 23.6 ± 8.2%, ΔCRE5 ~ 25.3 ± 6.6%, ΔCRE6 ~ 24.5 ± 10.2% and ΔFOXA1 (exon 1 and intron 1) ~ 87.4 ± 8.8% reduction relative to basal levels) (Fig. 4c, Supplementary Fig. 5a–f). In contrast, deletion of several negative control regions within the same TAD did not

significantly reduce *FOXA1* mRNA level (Fig. 4c, Supplementary Fig. 5g–i). Similar results were observed in 22Rv1 prostate cancer cells (Fig. 4d). As each clone expressed Cas9 protein at different levels, there may be a difference between genome editing efficiencies between the clones. We compared the CRISPR/Cas9

**Fig. 4 Functional dissection of putative *FOXA1* CREs. a** Representative western blot probed against Cas9 in LNCaP clones ($n = 5$ independent clones) derived to stably express Cas9 protein upon blasticidin selection. **b** Representative western blot probed against Cas9 in 22Rv1 clones ($n = 4$ independent clones) derived to stably express Cas9 protein upon blasticidin selection. **c** *FOXA1* mRNA expression normalized to housekeeping *TBP* mRNA expression upon CRISPR/Cas9-mediated deletion of each CRE using LNCaP clones ($n = 5$ independent experiments, each dot represents an independent clone). **d** *FOXA1* mRNA expression normalized to housekeeping *TBP* mRNA expression upon CRISPR/Cas9-mediated deletion of each CRE using 22Rv1 clones ($n = 4$ independent experiments, each dot represents an independent clone). **e** Representative western blot probed against Cas9 in LNCaP clones ($n = 4$ independent clones) derived to stably express the dCas9-KRAB fusion protein upon blasticidin selection. **f** Representative western blot probed against Cas9 in 22Rv1 clones ($n = 4$ independent clones) derived to stably express dCas9-KRAB fusion protein upon blasticidin selection. **g** *FOXA1* mRNA expression normalized to housekeeping *TBP* mRNA expression upon dCas9-KRAB-mediated repression of each CRE using LNCaP clones ($n = 4$ independent experiments, each dot represents an independent clone). **h** *FOXA1* mRNA expression normalized to housekeeping *TBP* mRNA expression upon dCas9-KRAB-mediated repression of each CRE using 22Rv1 clones ($n = 4$ independent experiments, each dot represents an independent clone). *FOXA1* mRNA expression was normalized to basal *FOXA1* expression prior to statistical testing. Δ indicates CRISPR/Cas9-mediated deletion, **i** indicates dCas9-KRAB-mediated repression. Error bars indicate ± s.d. Student's *t* test, n.s not significant, *$p < 0.05$, **$p < 0.01$, ***$p < 0.001$.

on-target genome editing efficiency across the five LNCaP cell line-derived clones with the relative *FOXA1* mRNA levels, and indeed observe a significant inverse correlation across all CREs (Pearson's correlation $r = 0.49$, $p < 0.005$) (Supplementary Fig. 6a) and agreeing trends for each individual CRE (Supplementary Fig. 6b).

Complementary to our findings using the wild-type CRISPR/Cas9 system, we next generated four LNCaP and four 22Rv1 cell line-derived dCas9-KRAB fusion protein expressing clones (Fig. 4e, f). Transient transfection of the same gRNAs used in the wild-type Cas9 experiments, targeting the six *FOXA1* plexus CREs (Supplementary Data) into our dCas9-KRAB LNCaP clones significantly decreased *FOXA1* expression relative to basal levels (iCRE1 ~ 24.6 ± 6.2%, iCRE2 ~ 42.2 ± 10.8%, iCRE3 ~ 25.3 ± 9.2%, iCRE4 ~ 23.3 ± 4.3%, iCRE5 ~ 30.2 ± 3.4%, and iCRE6 ~ 23.1 ± 8.1% reduction). Similarly, gRNAs targeting the dCas9-KRAB fusion protein to *FOXA1* decreased its expression (i*FOXA1* ~ 81.6 ± 11.8% reduction; Student's *t* test, $p < 0.05$, Fig. 4g). Analogous results were also observed in our four clonal 22Rv1 dCas9-KRAB cell lines (Student's *t* test, $p < 0.05$, Fig. 4h). Collectively, our results suggest that the six CREs control *FOXA1* expression.

We further assessed the regulatory activity of the six *FOXA1* plexus CREs by testing the consequent mRNA expression on other genes within the same TAD, namely *MIPOL1* and *TTC6*. ΔCRE1 and ΔCRE2 significantly reduced *MIPOL1* mRNA expression by ~ 38.4 ± 6.4% and ~ 48.4 ± 9%, respectively, relative to basal levels, whereas deletion of the other four CREs did not result in any significant *MIPOL1* expression changes (Student's *t* test, $p < 0.05$, Supplementary Fig. 7a). On the other hand, deletion of CREs each significantly reduced *TTC6* mRNA expression relative to its basal levels (ΔCRE1 ~ 52.9% ± 6.4%, ΔCRE2 ~ 66 ± 11.3%, ΔCRE3 ~ 55.5 ± 12.8%, ΔCRE4 44.9 ± 10.6%, ΔCRE5 43.1 ± 11.9% and ΔCRE6 52.2 ± 7.3% reduction (Student's *t* test, $p < 0.05$, Supplementary Fig. 7b), in agreement with the fact that *TTC6* shares its promoter with *FOXA1* as both genes are transcribed on opposing strands (Supplementary Fig. 7c).

Reduction in *FOXA1* mRNA expression resulting from the deletion of *FOXA1* plexus CREs may also impact gene expression downstream of *FOXA1*. We assessed the mRNA expression of several FOXA1 target genes, namely *SNAI2*, *ACPP*, and *GRIN3A*. Deletion of CREs resulted in significant change in *SNAI2* (upregulation; ΔCRE1 ~ 190%, ΔCRE2 ~ 162.8%, ΔCRE3 ~ 147.5%, ΔCRE4 ~ 133.3%, ΔCRE5 ~ 137.3%, ΔCRE6 ~ 120.8%, Δ*FOXA1* ~ 266.7%), *ACPP* (downregulation; ΔCRE1 ~ 73.5%, ΔCRE2 ~ 62.5%, ΔCRE3 ~ 69.6%, ΔCRE4 ~ 75.6%, ΔCRE5 ~ 70.9%, ΔCRE6 ~ 74.6%, Δ*FOXA1* ~ 52.2%) and *GRIN3A* expression (upregulation; ΔCRE1 ~ 138.2%, ΔCRE2 ~ 168.8%, ΔCRE3 ~ 144.6%, ΔCRE4 ~ 132.1%, ΔCRE5 ~ 131.4%, ΔCRE6 ~ 127%,

Δ*FOXA1* ~ 228%) (Student's *t* test, $p < 0.05$, Supplementary Fig. 7d–f). Collectively, our results support the regulation of most *FOXA1* plexus CREs towards *FOXA1* and its target genes.

**FOXA1 CREs collaborate to regulate its expression**. Expanding on the idea that multiple CREs can converge to regulate the expression of a single target gene[31,32,42], we asked whether the CREs we identified collaboratively regulate *FOXA1* mRNA expression. Here, we applied a transient approach that delivers Cas9 protein:gRNA as a ribonucleoprotein (RNP) complex formed prior to transfection that would avoid the heterogeneity of Cas9 protein expression across the prostate cancer cell clones (See Methods). We first validated this system through single CRE deletions, where we transiently transfected a set of gRNA targeting the CRE of interest. In accordance with data from our prostate cancer cell clones stably expressing wild-type Cas9 and dCas9-KRAB, individual CRE deletion resulted in a significant reduction in *FOXA1* mRNA expression: (ΔCRE1 ~ 29.3 ± 7.3%, ΔCRE2 ~ 36 ± 11.8%, ΔCRE3 ~ 30.6 ± 12.7%, ΔCRE4 ~ 24.5 ± 6.1%, ΔCRE5 ~ 23.7 ± 13.2%, ΔCRE6 ~ 26.8 ± 14.2% and Δ*FOXA1* ~ 96.2 ± 1.4% reduction (Student's *t* test, $p < 0.05$, Fig. 5a, Supplementary Fig. 8a–f). Next for combinatorial deletions, we prioritized the CREs that harbor more than one SNV(i.e, CRE1, CRE2, CRE4), and transiently transfected RNP complexes that target both CREs in various combinations (i.e., CRE1 + CRE2, CRE1 + CRE4, CRE2 + CRE4), and assessed *FOXA1* mRNA expression. Compared to negative control regions, the combinatorial deletion of ΔCRE1 + ΔCRE2, ΔCRE1 + ΔCRE4, and ΔCRE2 + ΔCRE4 resulted in a significant ~ 48.5 ± 4.5%, ~ 50.4 ± 2.9%, and ~ 45.2 ± 5.5% reduction in *FOXA1* mRNA expression, respectively (Student's *t* test, $p < 0.05$, Fig. 5b, Supplementary Fig. 9a–f) a fold reduction greater than single CRE deletions (Student's *t* test, Supplementary Fig. 10, $p < 0.05$). These results together demonstrate that these CREs collaboratively contribute to the establishment and regulation of *FOXA1* expression in prostate cancer.

**Disruption of FOXA1 CREs reduces prostate cancer cell growth**. As *FOXA1* is essential for prostate cancer growth (Fig. 1b–e), we next sought to assess the importance of the six FOXA1 plexus CREs towards prostate cancer cell growth. We adapted a lentiviral-based approach that expressed both the Cas9 protein and two gRNA that target each CRE for deletion (See Methods). Upon lentiviral transduction with subsequent selection, we separated LNCaP prostate cancer cells for RNA, DNA, and for cell proliferation. We first tested the system by measuring *FOXA1* mRNA expression, and independently observed significant reductions of *FOXA1* mRNA expression (ΔCRE1 ~ 18%, ΔCRE2 ~ 30%, ΔCRE3 ~ 15%, ΔCRE4 ~ 12%, ΔCRE5 ~ 35%,

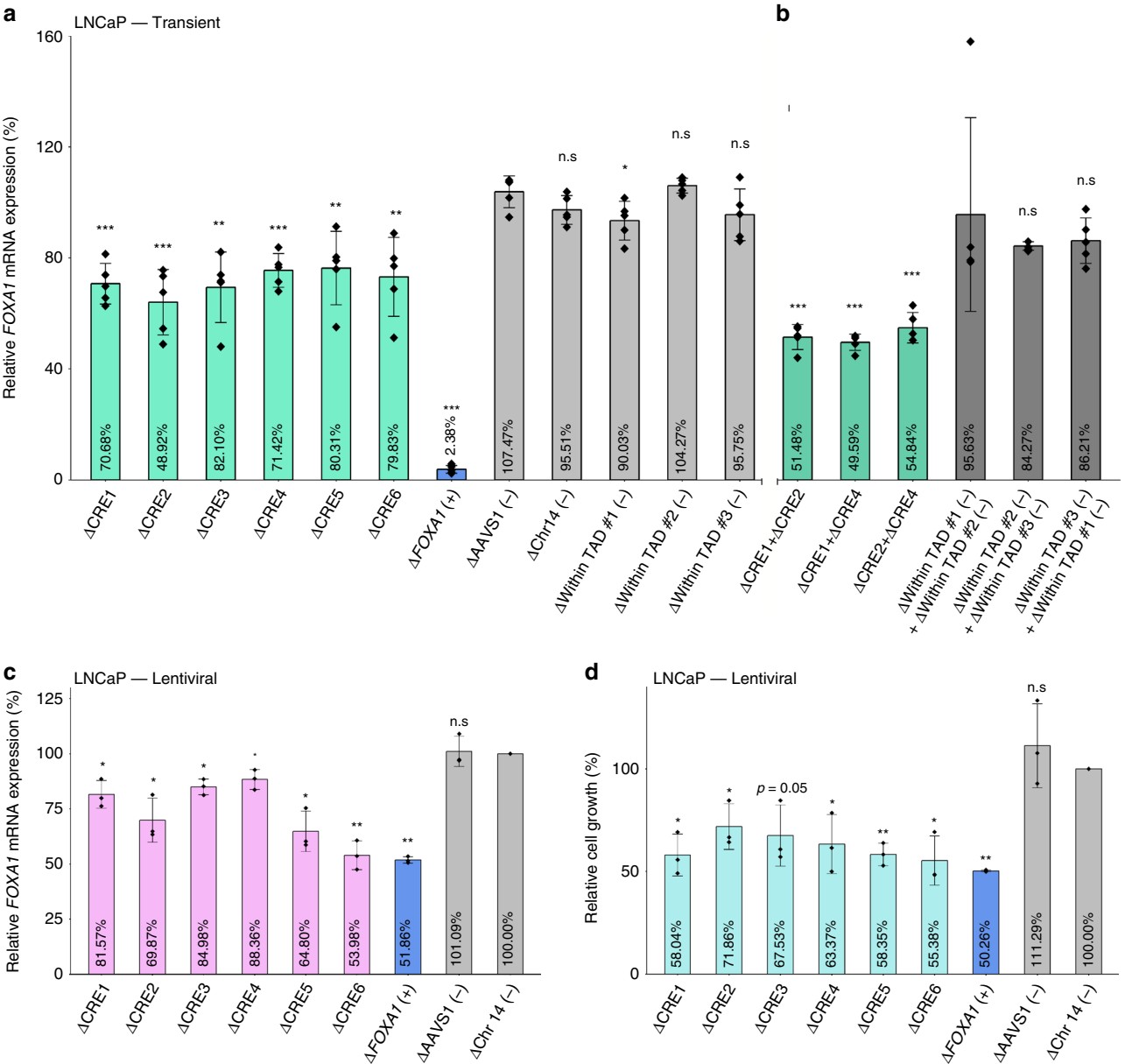

**Fig. 5 *FOXA1* CREs collaborate to regulate its expression and are critical for prostate cancer cell proliferation. a** *FOXA1* mRNA expression normalized to housekeeping *TBP* mRNA expression upon transient transfection-based CRISPR/Cas9-mediated deletion of CRE1, CRE2, CRE4, and sequential deletion combinations ($n = 5$ independent experiments). **b** *FOXA1* mRNA expression normalized to housekeeping *TBP* mRNA expression upon bulk lentiviral-based CRISPR/Cas9-mediated deletion of each CRE in LNCaP cells ($n = 3$ independent experiments). **c** Cell proliferation assay conducted after puromycin and blasticidin selection for LNCaP cells carrying deleted region of interest. Data were based on cell counting 6 days after seeding post-selection ($n = 3$, representative of three independent experiments). *FOXA1* mRNA expression upon deletion was normalized to basal *FOXA1* expression prior to statistical testing. *FOXA1* mRNA expression was normalized to basal LNCaP *FOXA1* expression prior to statistical testing. Δ indicates CRISPR/Cas9-mediated deletion. Error bars indicate ± s.d. Student's *t* test, n.s not significant, *$p < 0.05$, **$p < 0.01$, ***$p < 0.001$.

ΔCRE6 ~ 46%, and Δ*FOXA1* (exon 1 and intron 1) ~ 48% reduction (Student's *t* test, $p < 0.05$, Fig. 5c, Supplementary Fig. 11a–f). We then seeded these cells at equal density. Six days post seeding, we harvested the cells and observed a significant reduction in cell growth upon deleting any of the six *FOXA1* plexus CREs (ΔCRE1 ~ 42%, ΔCRE2 ~ 28%, ΔCRE3 ~ 33%, ΔCRE4 ~ 27%, ΔCRE5 ~ 42%, ΔCRE6 ~ 44% and Δ*FOXA1* (exon 1 and intron 1) ~ 50% reduction (Student's *t* test, $p < 0.05$, Fig. 5d). These results suggest that the six *FOXA1* plexus contribute to prostate cancer etiology, in agreement with their ability to regulate *FOXA1* expression and the essentiality of this gene in prostate cancer cell growth.

**SNVs mapping to *FOXA1* CREs can alter their activity.** Single-nucleotide variants can alter the transactivation potential of CREs[32,43–51]. In total, we found 10 SNVs called from 9 out of the 200 tumors that map to the six *FOXA1* plexus CREs (Fig. 6a, Supplementary Data). To assess the impact of these noncoding SNVs, we conducted luciferase assays comparing differential reporter activity between the variant and the wild-type allele of each CRE (Fig. 6b–k). We found that the variant alleles of 6 of the 10 SNVs displayed significantly greater luciferase reporter activity when compared with the wild-type alleles (Mann–Whitney *U* test, $p < 0.05$). Specifically, we observed the following fold-changes: chr14:37,887,005 A > G (1.65-fold),

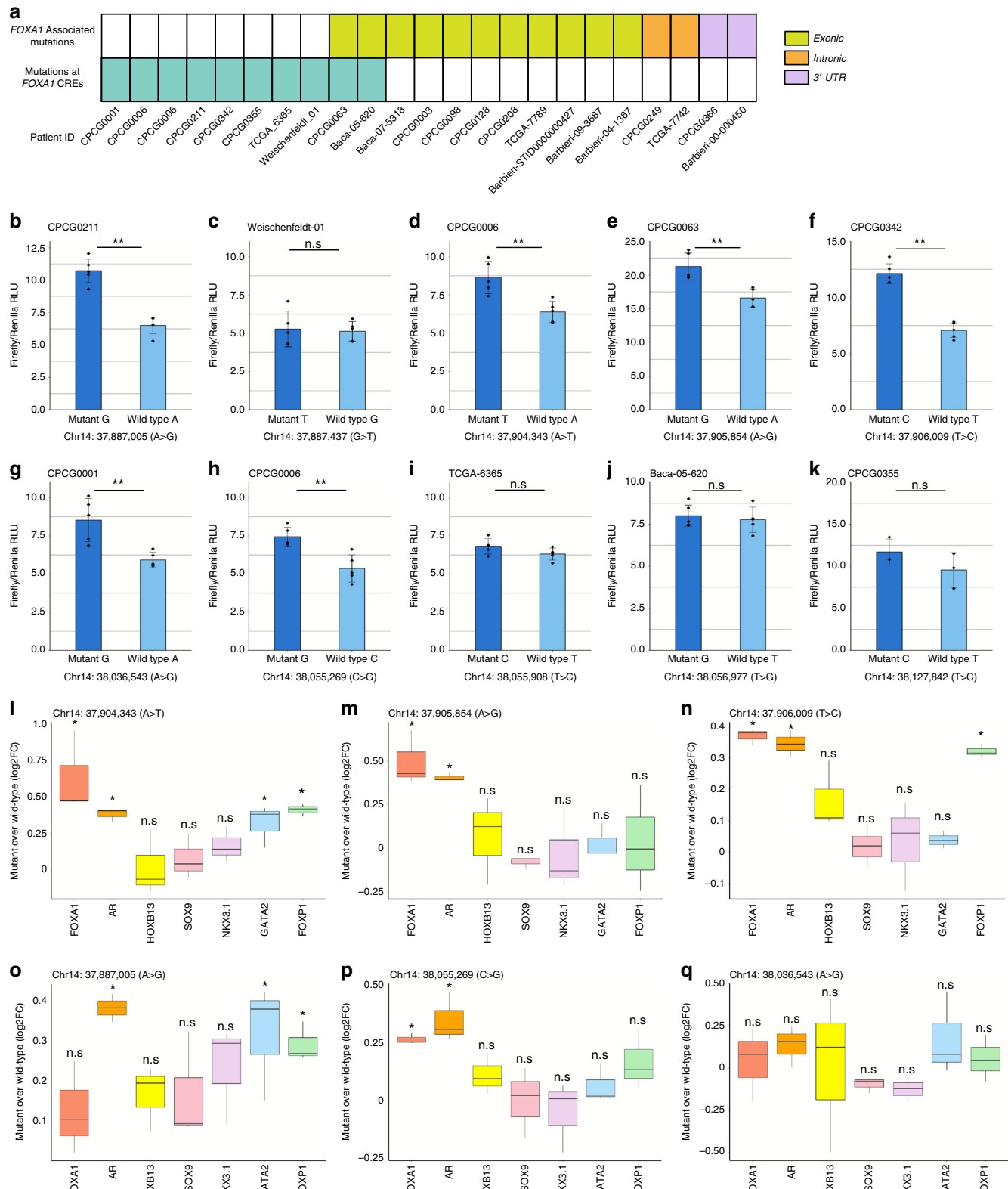

**Fig. 6 Subset of noncoding SNVs mapping to the *FOXA1* CREs are gain-of-function. a** Matrix showcasing the patients from the Fraser et al. data set[6] that harbor SNVs at the *FOXA1* CREs, exons, introns, and the 3'UTR of *FOXA1*. **b–k** Luciferase assays conducted in LNCaP cells. Barplot showcases the mean firefly luciferase activity normalized by *renilla* luciferase activity, error bars indicate ± s.d. $n = 5$ independent experiments ($n = 3$ for chr14: 38,127,842 (T > C)), and each diamond represents an independent experiment. Mann–Whitney $U$ test *$p < 0.05$, **$p < 0.01$. RLU = Relative Luciferase Unit. **l–q** Allele-specific ChIP-qPCR conducted on plasmids carrying the wild-type or variant sequence upon transient transfection in prostate cancer cells. Data is presented as $\log_2$ fold-change of variant sequence upon comparing to wild-type sequence ($n = 3$ independent experiments per ChIP). Student's $t$ test, n.s not significant, *$p < 0.05$, **$p < 0.01$, ***$p < 0.001$.

chr14:37,904,343 A > T (1.35-fold), chr14:37,905,854 A > G (1.28-fold), chr14:37,906,009 T > C (1.71-fold), chr14:38,036,543 A > G (1.44-fold), chr14:38,055,269 C > G (1.39-fold) (Fig. 6b–h). These results indicate that these SNVs can alter the transactivation potential of *FOXA1* plexus CREs in prostate cancer cells.

**SNVs mapping to *FOXA1* CREs can modulate the binding of TFs**. We next assessed if the changes in transactivation potential induced by noncoding SNVs related to changes in TF binding to CREs by allele-specific ChIP-qPCR[32,44,51] in LNCaP prostate cancer cells. We observed differential binding of FOXA1, AR, HOXB13, GATA2, and FOXP1 for the chr14:37887005 (A > G) SNV found in CRE1; the chr14:37904343 (A > T), chr14:37905854 (A > G), and chr14:37906009 (T > C) SNVs found in CRE2; and the chr14:38055269 (C > G) SNV found in CRE4 (Student's *t* test, *p* < 0.05, Fig. 6l–p). In contrast, SOX9 and NKX3.1 binding was unaffected by these SNVs (Fig. 6l–q). Compared with the wild-type sequence, chr14:37,887,005 A > G significantly increased AR binding (1.31-fold increase), GATA2 binding (1.25-fold increase) and FOXP1 binding (1.23-fold increase); chr14:37,904,343 A>T significant increased AR binding (1.30-fold increase), GATA2 (1.25-fold increase) and FOXP1 (1.33-fold increase); chr14:37,905,854 A > G significantly increased FOXA1 binding (1.41-fold increase) and AR binding (1.33-fold increase); chr14:37,906,009 T > C significantly increased the binding of FOXA1 (1.29-fold increase), AR (1.31-fold increase), HOXB13 (1.13-fold increase) and FOXP1 (1.25-fold increase); and chr14:38,055,269 C > G significantly increased FOXA1 binding (1.20-fold increase). Notably, all six SNVs increased the binding of the TFs known to bind at these CREs. In contrast, none of the SNVs significantly decreased the binding of these TFs. Our observations suggest that gain-of-function SNVs populate the *FOXA1* plexus CREs.

## Discussion

Modern technologies and understanding of the epigenome allow the possibility of probing CRE(s) involved in regulating genes implicated in disease. Despite *FOXA1* being recurrently mutated[5–8,11] and playing potent oncogenic roles in prostate cancer etiology[9,10,13], the CREs involved in its transcriptional regulation are poorly understood. Understanding how *FOXA1* is expressed can provide a complementary strategy to antagonize FOXA1 in prostate cancer.

We used the DHS profiled in prostate cancer cells to identify putative *FOXA1* CREs by annotating these regions with five different histone modifications, TF-binding sites and noncoding SNVs profiled in prostate cancer cells and primary prostate tumors. Our efforts identified and validated a set of six active CREs involved in *FOXA1* regulation, agreeing with a recent report where a subset of our CREs map to loci suggested to be in contact with the *FOXA1* promoter[52]. The disruption of these six distal CREs each significantly reduced *FOXA1* mRNA levels, similar to what has been demonstrated for *ESR1* in luminal breast cancer[32], *MLH1* in Lynch syndrome[53], *MYC* in lung adenocarcinoma and endometrial cancer[54] and *AR* in mCRPC[55,56]. Through combinatorial deletion of two CREs, *FOXA1* mRNA levels were further reduced in comparison with single CRE deletions, raising the possibility of CRE additivity[57]. The deletion of the *FOXA1* plexus CREs also significantly reduced prostate cancer cell proliferation at levels comparable to what has been reported upon deletion of the amplified CRE upstream of the *AR* gene in mCRPC[55], suggestive of onco-CREs as reported in lung[54], and prostate[55] cancer.

More than 90% of SNVs found in cancer map to the noncoding genome[58,59] with a portion of these SNVs mapping to CREs

altering their transactivation potential[32,44–46] and/or downstream target gene expression[48,58,60]. We extended this concept with SNVs identified from primary prostate tumors mapping to *FOXA1* plexus CREs. We observed that a subset of these SNVs can alter transactivation potential by modulating the binding of specific TFs whose cistromes are preferentially burdened by SNVs in primary prostate cancer[59]. Our findings complement recent reports of SNVs found in the noncoding space of *FOXA1* that could affect its expression[14,61]. The *FOXA1* plexus CREs we identified here are also reported to be target of structural variants in both the primary and metastatic settings[9,62], including tandem duplication in ~ 14% (14/101) mCRPC tumors over CRE2[62], amplification, duplication, and translocation over CRE3, 4, 5[9]. Notably, the translocation and duplication defining the FOXMIND enhancer driving *FOXA1* expression reported in primary and metastatic settings harbors the CRE3 element we characterized[9]. Collectively, these studies combined with our discoveries reveal the fundamental contribution of the *FOXA1* plexus in prostate cancer etiology. As a whole, our findings in conjunction with recent reports suggest that CREs involved in the transcriptional regulation of *FOXA1* may be hijacked in prostate tumors through various types of genetic alterations.

Despite initial treatment responses from treating aggressive primary and metastatic prostate cancer through castration to suppress AR signaling[4], resistance ensues as 80% of mCRPC tumors harbor either *AR* gene amplification, amplification of a CRE upstream of *AR*, or activating *AR* coding mutations[11,55,62]. Given the AR-dependent[15,18] and AR-independent[25] oncogenic activity of FOXA1 in prostate cancer, its inhibition is an appealing alternative therapeutic strategy. Our dissection of the *FOXA1* cis-regulatory landscape complement recent findings by revealing loci that are important for the regulation of *FOXA1*. Theoretically, direct targeting of the CREs regulating *FOXA1* would downregulate *FOXA1* levels and could therefore serve as a valid alternative to antagonize its function.

Taken together, we identified *FOXA1* CREs targeted by SNVs that are capable of altering transactivation potential through the modulation of key prostate cancer TFs. The study supports the importance of considering CREs not only as lone occurrences but as a team that work together to regulate their target genes, particularly when considering the impact of genetic alterations. As such, our work builds a bridge between the understanding of *FOXA1* transcriptional regulation and new routes to FOXA1 inhibition. Aligning with recent reports[9,10,13], our findings support the oncogenic nature of *FOXA1* in prostate cancer. Gaining insight on the cis-regulatory plexuses of important genes such as *FOXA1* in prostate cancer may provide new avenues to inhibit other drivers across various cancer types to halt disease progression.

## Methods

**Cell culture**. LNCaP and 22Rv1 cells were cultured in RPMI medium, and VCaP cells were cultured in DMEM medium, both supplemented with 10% FBS, and 1% penicillin–streptomycin at 37 °C in a humidified incubator with 5% $CO_2$. These prostate cancer cells originated from ATCC. 293FT cells were purchased from ThermoFisher Scientific (Cat. No. R70007) maintained in complete DMEM medium (DMEM with 10% FBS (080150, Wisent), L-glutamine (25030–081, ThermoFisher) and non-essential amino acids (11140–050, ThermoFisher) supplemented with 50 mg/mL Geneticin (4727894001, Sigma-Aldrich). The cells are regularly tested for Mycoplasma contamination. The authenticity of these cells was confirmed through short tandem repeat profiling.

**Prostate tumors and cancer cell lines expression**. Cancer cell line mRNA abundance data were collected from the Cancer Dependency Map Project (DEPMAP; https://depmap.org/portal/; RNA-seq TPM values from 2018q4 version)[63] projects. Prostate tumor mRNA abundance data were collected from The Cancer Genome Atlas (TCGA) prostate cancer (TCGA-PRAD) project via the Xena

Browser (https://xenabrowser.net/; data set description: TCGA prostate adeno-carcinoma (PRAD) gene expression by RNA-seq (polyA+Illumina HiSeq; RSEM)).

**Prostate cancer cell line gene essentiality**. Essentiality scores were collected from the Cancer Dependency Map Project (DEPMAP) (https://depmap.org/portal/download/; dataset description: 2018q4 versions of the "cell line metadata" and "combined RNAi"[64], and all five non-cancer cell lines were removed (cell lines where the "Primary Disease" was listed in the metadata as one of the following: fibroblast, immortalized, immortalized_epithelial, non-cancerous, primary cells, or unknown). To compare gene essentiality between prostate cancer cell lines and others, essentiality scores for *FOXA1* were collected from all available cell lines ($n = 707$). To perform a permutation test, the median of eight randomly selected cell lines was calculated one million times to generate a background distribution of essentiality scores across all cell types available. The median essentiality score from the eight prostate cancer cell lines was calculated and its percentile within the background distribution is reported.

**siRNA knockdown and cell proliferation assay**. In all, 300,000 LNCaP cells (Day 0) were reverse transfected with siRNA (siFOXA1 using Lipofectamine RNAimax reagent (ThermoFisher Scientific, Cat. No. 13778150)). Cells were counted using Countess automated cell counter (Invitrogen). Whole-cell lysates LNCaP cells after siRNA-mediated *FOXA1* knockdown were collected at 96 h post transfection in RIPA buffer. Protein concentrations were determined through the bicinchoninic acid method (ThermoFisher Scientific, Cat. No. 23225). Then 25 μg of lysate was subjected to SDS-PAGE. Upon completion of SDS-PAGE, protein was transferred onto PVDF membrane (Bio-Rad, Cat. No. 1704156). The membrane was blocked with 5% non-fat milk for 1 h at room temperature with shaking. After blocking, anti-FOXA1 (Abcam Cat. No. 23737) in 2.5% non-fat milk was added, and was incubated at 4 °C overnight. Next day, the blot was washed and incubated with IRDye 800CW Goat Anti-Rabbit IgG secondary antibody (LI-COR, Cat. No. 925–32211) at room temperature for 1 h. The blot was then washed and assessed with the Odyssey CLX imaging system (LI-COR).

**Identifying putative *FOXA1* CREs**. Putative *FOXA1* CREs were identified through the use of Cross Cell-Type Correlation based on DNase I Hypersensitivity (C3D) (https://github.com/tahmidmehdi/C3D)[37]. Predicted interacting DNase I Hyper-sensitivity Sites (DHS) with a Pearson's correlation above 0.7[65] were kept for downstream analysis.

**Hi-C and TADs in LNCaP cells**. Hi-C and TADs conducted and called, respectively, in LNCaP cells is publicly available off ENCODE portal (ENCSR346DCU). Visualization of the Hi-C dataset is available on the Hi-C Browser (http://promoter.bx.psu.edu/hi-c/)[66].

**Clonal wild-type Cas9 and dCas9-KRAB-mediated validation**. Lentiviral parti-cles were generated in 293FT cells (ThermoFisher) using the pMDG.2 and psPAX2 packaging plasmids (Addgene; #12259 and #12260, a gift from Didier Trono) alongside the Lenti-Cas9-2A-Blast plasmid (Addgene #73310, gift from Jason Moffat) and collected 72 hrs post transfection. LNCaP and 22Rv1 cells were then transduced for 24–48 h with equal amounts of virus followed by selection with media containing blasticidin (7.5 μg/mL for LNCaP cells, 6 μg/mL for 22Rv1 cells). Upon selection, clones were derived by serial dilution with subsequent single cell seeding into 96-well plates containing selection media. Cas9 protein expression for each clone was then assessed through Western blotting (1° Ms-Cas9 (Cell Sig-nalling Technology, Cat. No. #14697) 1:1000, Ms-GAPDH 1:5000 (Santa Cruz Biotechnology, Cat. No. #sc47724) in 5% non-fat milk; 2° HRP-linked Anti-Mouse IgG (Cell Signalling Technology, Cat. No. #7076 S) 1:10,000 in 2.5% non-fat milk). The full unprocessed blot is in the Source Data File.

Lentiviral particles were generated in 293FT cells (ThermoFisher) using the pMDG.2 and psPAX2 packaging plasmids (Addgene; #12259 and #12260, a gift from Didier Trono) alongside the Lenti-dCas9-KRAB-blast plasmid (Addgene #89567, a gift from Gary Hon) and collected 72 hrs post transfection. LNCaP and 22Rv1 cells were then transduced for 24–48 h with equal amounts of virus followed by selection with media containing blasticidin (7.5 μg/mL for LNCaP cells, 6 μg/mL for 22Rv1 cells). Upon selection, clones were derived by serial dilution with subsequent single cell seeding into 96-well plates containing selection media. dCas9-KRAB protein expression for each clone was then assessed through western blotting (1° Ms-Cas9 (Cell Signalling Technology, Cat. No. #14697) 1:1000, Ms-GAPDH 1:5000 (Santa Cruz Biotechnology, Cat. No. #sc47724) in 5% non-fat milk; 2° HRP-linked Anti-Mouse IgG (Cell Signalling Technology, Cat. No. #7076 S) 1:10,000 in 2.5% non-fat milk). The full unprocessed blot is in the Source Data File.

For gRNA design, five to six unique crRNA molecules (Integrated DNA Technologies) were designed to tile across the region of interest using the CRISPOR tool (http://crispor.tefor.net/)[67] and the Zhang lab CRISPR Design tools (http://crispr.mit.edu/)[68] (Supplementary Data). Each CRISPR RNA (crRNA) and tracrRNA (Integrated DNA Technologies) were duplexed according to company supplier protocol to a concentration of 50 μM. Upon generation of the clones, six guides (crRNA-tracrRNA duplexes) for each region of interest were pooled into a single tube (1 μL each guide, 6 μL per reaction) (Integrated DNA Technologies).

At last, 1 μL (100 μM) of electroporation enhancer (Integrated DNA Technologies) was added to the mix (7 μL total) prior to transfection. The entire transfection reaction was transfected into 350,000 cells through Nucleofection (SF Solution EN120—4D Nucleofector, Lonza). Cells were then harvested 24 h post transfection for RNA and DNA for RT-PCR and confirmation of deletion, respectively.

**Transient Cas9-mediated disruption of CREs**. Deletion of elements through this method were achieved through the transfection of Cas9 nuclease protein com-plexed with the crRNA (Integrated DNA Technologies). In brief, five to six unique crRNA molecules (Integrated DNA Technologies) were designed to tile across the region of interest using the CRISPOR tool (http://crispor.tefor.net/)[67] and the Zhang lab CRISPR Design tools (http://crispr.mit.edu/)[68]. Each crRNA and tracrRNA (Integrated DNA Technologies) were duplexed according to company supplier protocol to a concentration of 50 μM. The six crRNA-tracrRNA duplexes were pooled into a single tube (6 μL per reaction), prior to adding 1 μL (5 μg) of Alt-R S.p HiFi Cas9 Nuclease 3NLS (Integrated DNA Technologies). The reaction was incubated at room temperature for 10 min for RNP complex formation. At last, 1 μL (100 μM) of electroporation enhancer (Integrated DNA Technologies) was added to the mix prior to transfection. The entire transfection reaction was transfected into 350,000 cells through Nucleofection (SF Solution EN120–4D Nucleofector, Lonza). Cells were then harvested 24 h post transfection for RNA and DNA for RT-PCR and confirmation of deletion, respectively. For double deletions, two sets of guide RNA-RNP complex (10 μg of Alt-R S.p HiFi Cas9 Nuclease 3NLS) were transfected and harvested 24 h post transfection for RNA and DNA for RT-PCR and confirmation of deletion, respectively. To control for double deletions, two negative control regions within the TAD were also compounded (Supplementary Data).

**RT-PCR assessment of gene expression upon deletion of CREs**. DNA and RNA were harvested with Qiagen AllPrep RNA/DNA Kit (Qiagen, Cat. No. 80204). Next, cDNA was synthesized from 300 ng of RNA using SensiFast cDNA Synthesis kit (Bioline, Cat. No. BIO-65054), and mRNA expression levels for various genes of interest were assessed. The primer sequences for expression evaluation are in Supplementary Data. Differential gene expression was calculated upon normal-ization with *TBP* (housekeeping gene). Statistical significance was calculated using Student's *t* test in R.

**Confirmation of Cas9-mediated deletion of CREs**. Deletion of CREs were con-firmed through PCR amplification of the intended region for deletion, followed by the T7 endonuclease assay (Integrated DNA Technology). Primer sequences used for PCR amplification are in Supplementary Data. PCR products were then loaded onto a 1% agarose gel for visualization. The agarose gel to assess the on-target genome editing efficiency was done through densitometry using ImageJ. The correlation between on-target genome editing efficiency and *FOXA1* mRNA expression reduction was drawn through Pearson's correlation in R.

**Cell proliferation upon deletion of *FOXA1* CREs**. Pairs of gRNAs flanking the CREs of interest, *FOXA1* promoter and control regions were designed using CRISPOR (http://crispor.tefor.net/) and Zhang lab CRISPR Design tool (http://crispr.mit.edu/) (Supplementary Data). Each pair of gRNAs were cloned into the lentiCRISPRv2 (Addgene; a gift from Feng Zhang #52961) and the lentiCRISPRv2-Blast (Addgene; a gift from Feng Zhang #83480) plasmid as pre-viously described[69]. Lentiviral particles were generated in 293FT cells (Thermo-Fisher) using the pMDG.2 and psPAX2 packaging plasmids (Addgene; #12259 and #12260, a gift from Didier Trono), and collected 72 hrs post transfection. LNCaP cells were transduced for 24–48 hrs with equal amounts of virus, followed by selection with media containing puromycin (3.5 μg/mL, ThermoFisher) and blas-ticidin (7 μg/mL, Wisent). Cells were harvested upon selection for RNA and DNA for RT-PCR and confirmation of DNA cleavage, respectively. For cell proliferation, cells were seeded at equal density per well (on a 96-well plate; Day 1) upon puromycin and blasticidin selection. Growth of the cells were monitored by cell counting using Countess automated cell counter (Invitrogen). Cell numbers were calculated as a percentage compared to negative control. Statistical significance was calculated using Student's *t* test.

**Luciferase reporter assays**. Each region of interest was ordered as gBlocks from Integrated DNA Technologies. The regions were cloned into the BamHI restriction enzyme digest site of the pGL3 promoter plasmid (Promega). On Day 0, 90,000 LNCaP cells were seeded in 24-well plates. Next day (Day 1), pGL3 plasmids harboring the wild-type and variant sequences were co-transfected with the pRL *Renilla* plasmid (Promega) using Lipofectamine 2000. 48 h later, the cells were harvested, and dual luciferase reporter assays were conducted (Promega). Notably, inserts of both forward and reverse directions were tested using this assay as enhancer elements are known to be direction-independent. Final luminescence readings are reported as *firefly* luciferase normalized to *renilla* luciferase activity. The assessment of each mutation was conducted in five biological replicates. Sta-tistical significance was assessed by Mann–Whitney *U* test in R. gBlock sequences are in Supplementary Data.

**Allele-specific ChIP-qPCR**. In brief, pGL3 plasmids containing the wild-type sequence and the mutant sequence used in the luciferase reporter assay were transfected into 7 million cells (2 µg per allele, per one million cells) using Lipofectamine 2000 (ThermoFisher Scientific), per manufacturer's instructions. Next day, each antibody (FOXA1 5 µg, Abcam, ab23738; AR 5 µg, Abcam, ab1083241; HOXB13 5 µg, Abcam, ab201682; SOX9 5 µg, Abcam, ab3697; GATA2 5 µg, Abcam, ab22849; FOXP1 5 µg, Abcam, ab16645; NKX3.1 10 µl, Cell Signalling Technology, #83700) was conjugated with 10 µL of each Dynabeads A and G (ThermoFisher Scientific) for each ChIP for 6 h with rotation at 4 °C. When antibody beads conjugates were ready for use, cells were lifted using trypsin and fixed by resuspending with 300 µL of 1% formaldehyde in PBS for 10 min at room temperature. 2.5 M Glycine was added to quench excess formaldehyde (final concentration 0.125 M). Cells were then washed with cold PBS and lysed using 300 µL of Modified RIPA buffer (10 mM TrisHCl, pH 8.0; 1 mM EDTA; 140 mM NaCl; 1% Triton X-100; 0.1% SDS; 0.1% sodium deoxycholate) supplemented with protease inhibitor. The lysate was subject to 25 cycles of sonication (30 s ON 30 s OFF) using Diagenode Bioruptor Pico (Diagenode). In all, 15 µL of sonicated lysate was set aside as input with the rest used for chromatin pulldown through addition of antibody beads conjugates and overnight incubation at 4 °C with rotation. Next day, the beads were washed once with Modified RIPA buffer, washed once with Modified RIPA buffer + 500 mM NaCl, once with LiCl buffer (10 mM TrisHCl, pH 8.0; 1 mM EDTA; 250 mM LiCl; 0.5% NP-40; 0.5% sodium deoxycholate) and twice with Tris-ETDA buffer (pH 8). After wash, beads and input were decrosslinked by addition of 100 µL Decrosslinking buffer and incubation at 65 °C for 6 h. Samples were then purified and eluted. ChIP and input DNA were then used for allele-specific ChIP-qPCR using MAMA primers as described previously (Supplementary Data). Fold-change significance was calculated using Student's $t$ test in R.

All analyses were done using hg19 reference genome coordinates.

**Reporting summary**. Further information on research design is available in the Nature Research Reporting Summary linked to this article.

## Data availability

Genomic and Epigenomic data sets used to support this study can be found from the following accession codes: primary tumors—H3K27ac ChIP-seq (GSE96652), SNVs called from primary tumors (https://dcc.icgc.org/projects/PRAD-CA), FOXA1, AR, and HOXB13 ChIP-seq in primary prostate tumors is available under the following accession code: GSE137527 and EGAS00001003928, TF ChIP-seq data were from public databases of ReMap and ChIP-Atlas. All other relevant data supporting the key findings of this study are available within the article and its Supplementary Information files or from the corresponding author upon reasonable request. Data used to generate the figures are available in the Source Data file. A reporting summary for this Article is available as a Supplementary Information file.

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

## Acknowledgements

We thank the Princess Margaret Genomics Centre and the Princess Margaret Bioinformatics group for providing support and infrastructure for the computational analysis of this work as well as high-throughput sequencing support and the NKI Core Facility Molecular Pathology and Biobanking for technical support and providing tissue specimens. We thank all the members of the M.L. lab for their fruitful discussions and feedback. This work is supported Prostate Cancer Canada, Ontario Institute for Cancer Research funded by the Government of Ontario, The Princess Margaret Cancer Foundation (M.L. and R.G.B.), Movember Foundation (RS2014-04 to M.L. and RS2014-01 to P.C.B.), the Radiation Medicine Program Academic Enrichment Fund (R.G.B.), Terry Fox Research Institute New Investigator Award (P.C.B.), Canadian Institute of Health Research (CIHR) (FRN-153234 to M.L.), and New Investigator Award (P.C.B. and M.L.), Canadian Cancer Society Research Scientist Award (R.G.B.), Cancer Cancer Society Impact Award (P.C.B), the Dutch Cancer Society KWF/Alpe d'HuZes (10084 and NKI 2014-6711 ALPE to W.Z.), CIHR Graduate Scholarship—Master's Research Award (J.R. H., M.T.). CIHR Graduate Scholarship—Doctoral Research Award (P.M., J.T.H.), Canadian Breast Cancer Foundation (CBCF) postdoctoral fellowship (K.J.K.), Investigator Award from the Ontario Institute for Cancer Research (M.L. and P.C.B.) and Movember Rising Star Award from Prostate Cancer Canada (M.L. and P.C.B.).

## Author contributions

S.Z. and M.L. conceptualized the study. S.Z. designed and conducted most of the experiments with help from F.S., G.G., M.T., K.J.K., J.T.H., C.A., H.Y.Y., Y.Z. and S.C. J.R.H. implemented most of the computational analyses and statistical approaches with help from S.A.M., P.M., M.A., A.M., V.H., T.N.Y., S.M.G.E., T.M.S. and J.L. under the supervision of W.Z., T.v.d.K., T.J.P., M.F., P.C.B., R.G.B., H.H.H., or M.L. Figures were designed by S.Z. with assistance from J.R.H. and S.A.M. The manuscript was written by S.Z., J.R.H. and M.L. with assistance from all authors. M.L. oversaw the study.

## Competing interests

W.Z. receives project funding from Astellas Pharma but the components of this manuscript provided by W.Z. and his team were funded by KWF Dutch Cancer Society and Oncode Institute. The other authors declare no competing interests.
