## [Peer Review File · Nature Communications]

Reviewers' comments:

Reviewer #1 (Remarks to the Author):

Functional validation of genomic elements using CRISPR/Cas9 is a major task of current functional genomics studies. FOXA1 has been found to be essential for the growth of prostate cancer cells or tumorigenesis. This study intended to validate the functions of 6 cis-regulatory elements for FOXA1 expression in prostate cancer cells and the functional genetic variants in these cis-regulatory elements. There are several concerns of this study.

1. CRISPR/Cas9 was used to examine the functions of cis-regulatory elements. However, key descriptions of this method are missing, such as sequences and locations of crRNA in each cis-regulatory element; T7 endonuclease assays were used to examine the efficiency of CRISPR/Cas9, but this assay only detects heterozygous mutations, did authors examine homozygous mutations and how? Where are the primer sequences and locations? Clearly, CRISPR/Cas9-treated cells are mixed cell populations of mutations; should publish at the level of Nature Communications, single colony with homozygous mutations would be essential for this type of functional genetic assays.
2. This manuscript is lack of proper citations of previous studies in the field; e.g., siRNA knockdown of FOXA1 suppressed the growth of prostate cancer cells (Fig. 2D-F) was reported in several studies previously; cis-regulatory elements of FOXA1 have been investigated in several studies previously and feedforward regulation of FOXA1 was also discovered previously.
3. It is over-interpreted for the studies of functional genetic variants in the cis-regulatory elements. It is a correlation but not a functional study. CRISPR/Cas9 would help this part.
4. It is quite confusing for the relation between Fig. 1 and the rest of studies.

Reviewer #2 (Remarks to the Author):

In this manuscript, Zhou and colleagues identify putative cis-regulatory elements of FOXA1 and experimentally validated 6 of them that drive FOXA1 expression in prostate cancer (PCa) cells. Additionally, they report the presence of somatic single-nucleotide mutations at these elements in PCa patients, which functionally enhance their transactivation potential in reporter assays. Although these findings are intriguing, the genomic and functional data presented in this study is very preliminary and lacks important details and validations. The conclusion that deletion of the CREs decreases FOXA1 expression and cell proliferation is appropriate, but the experimental evidence supporting this conclusion can be improved upon. In addition, the experiments used to support the conclusion that the SNVs found in patients are activating mutations at CREs of FOXA1 are suboptimal. Thus, this study requires more functional experiments (with important controls) carried out in multiple PCa cells lines. For example, functional validation of the SNVs in vitro and in vivo would provide direct relevance of those mutations to cancer progression. Specific comments/questions are listed below:

Major:

- 1) In Fig2B, the authors predict several of the PCa transcription factors to bind to the putative cis-regulatory elements of FOXA1. However, they do not interrogate as to which one of these factors are required for their activation. This is an important question that remains unanswered in this study.
- 2) In fig. 4G-I, along with supp. fig. 5, the authors performed deletions of various CREs of FOXA1 using CRISPR/Cas9. However, it looks as though the efficiency of the deletions is highly variable (supp. Fig. 5a vs 5c). Thus, it is hard to tell how well the decrease in FOXA1 expression correlates with the CRE deletions as opposed to random other mutations generated by one or both sgRNAs targeting the flanking regions of the CREs. It would be stronger evidence that the deletion of those CREs decreases FOXA1 expression if the authors were able to estimate the deletion efficiency and correlate that with the reduction of expression. In addition, isolating monoclonal cells with successful

deletion of the CREs and assessing the FOXA1 expression in those cells compared to wild type cells can give more solid evidence for this relationship. This comment can be applied to the combination deletion experiment as well as the proliferation assays (4G-I).

3) The authors use a pair of sgRNAs to disrupt the CREs. This approach is very inefficient as it requires both the sgRNAs to be present inside the same cell for the deletion to occur, thus ensuing in only a minute fraction of the cells harboring a successful deletion. Additionally, many of the cells would simply harbor in-dels at the cut sites rather than deletions of the intermediate DNA (which is also validated by the T7 endonuclease assays - a deletion would not generate a mismatch). This is particularly true when sequentially targeting multiple CREs in the cells. Thus, the authors should corroborate the CRE-disruption data using orthogonal approaches, such as the dCas9-KRAB system.

4) Related to above, what is the effect of disrupting these CREs on other TAD genes, such as TTC6?

5) Additionally, are there other SNVs within this FOXA1 TAD genomic region? The author should report all the SNVs found in these region and test that non-enhancer SNVs are not functionally relevant in their reported assays.

6) It is possible that any genomic cut within the FOXA1 TAD could ensue in modest down-regulation of FOXA1. Thus, the authors should include an important control sgRNA pair that cuts within the TAD but not at a putative CRE element.

7) For the sequential disruption of CREs, were the negative control cuts also compounded? These targeted cutting within the FOXA1 TAD could simply result in collapsing with 3D structure of the TAD instead of the intended focused deletion of the CRE element alone. Thus, it is important to use intra-TAD negative control sgRNA pairs and sequentially compound them when deleting multiple CREs together.

8) Since CREs affect the expression of the target gene through chromatin looping, plasmid-based firefly luciferase assays are not the best approach to testing this phenomenon, due to the extragenomic nature of plasmids.

9) Most importantly, almost every experiment in this study is only carried out in the LNCaP cells. The authors need to use other AR/FOXA1+ PCa models.

10) The authors report the presence of SNVs within these cis-elements in patient tumors; however, the functional validation of the SNVs found in patients is weak. Are these event hetero or homozygous? Are they within known motifs of the TFs there were predicted to bind to these elements in the in-silico analyses?

11) What is the expression of FOXA1 in these patients relative to WT enhancer controls. Even though the incorporation of these SNVs seemed to increase expression, it may not be a result of chromatin looping. A better way to test this hypothesis is to use CRISPR/Cas9 with guide RNAs targeting the CREs in a cell model, isolate monoclonal cells with the mutations seen in patients, and observe the expression of FOXA1.

Reviewer #3 (Remarks to the Author):

In the present study, Zhou S. et al. identify 6 putative FOXA1 cis-regulatory elements using single nucleotide variant calls from whole genome sequencing data from primary prostate cancer samples, paired with publicly available DHS, Hi-C, and TADs datasets of the prostate cancer cell line. They further singly or sequentially delete these elements in LNCaP cells with CRISPR/Cas9 approaches, showing that these deletions result in the reduction of FOXA1 expression and cell growth. They also study the function of 10 single-nucleotide variants in these elements using luciferase reporter assays and find that 6 of 10 SNVs have gain-of-function activity.

Overall, this study addresses the regulation of FOXA1, a critical transcription factor, in prostate cancer, and includes multiple novel findings, including the first delineation of the complex network of cis-regulatory elements of FOXA1. The manuscript is well written and well presented. At the same time, there are several issues that could be addressed to improve the manuscript. These are described below:

1) The authors screened for somatic SNVs from whole genome sequences of 200 primary prostate cancers, thus identifying 10 total SNVs in FOXA1 CREs. Given that the frequency of genomic alterations is significantly higher in metastatic prostate cancer compared to primary disease, could the authors provide a genomic analysis using whole genome sequencing data from metastatic castration-resistant prostate cancer (mCRPC)? Recently, there have been several different whole genome sequencing studies published in mCRPC, including several in Cell, and a study from the Netherlands recently reported on BioRxiv. I realize that it would be beyond the scope of this project to functionally characterize additional SNVs found with this analysis, but it would be good to understand how many more FOXA1s would be nominated based on SNV calls from metastatic disease.

2) The authors show that the nominated SNVs affect the transactivation potential of CREs via luciferase assays. Do any of these SNVs have a large enough effect such that there is an impact on cell growth/proliferation? (i.e., the one or two with the highest reporter activity?)

3) As a complementary strategy to assess gain of function of FOXA1 based on the nominated CREs (or the SNVs with the CREs), have the authors check the downstream genes regulated by FOXA1?

4) As the authors mentioned, FOXA1 promotes prostate cancer progression partially through AR signaling. Have the authors check the AR signaling when disrupting their nominated CREs?

5) The putative FOXA1 CRE1 and CRE2 are located in the gene MIPOL1. Have the authors check the impact on MIPOL1 when using CRISPRn to delete CRE1/2?

Minor issues include the following:

6) In Figure 1, what are the FOXA1 mRNA levels with the siRNA knockdown?

7) The authors use CRISPRn to delete CREs. Have the authors quantified the deletion efficiency beside the T7 assay shown in Supplemental Figure 5? Will the different efficiencies impact the FOXA1 expression levels and cell growth in Figure 4?

8) Perhaps I missed these, but I think the supplementary tables are missing.

Overall, this is a provocative study with novel findings; the manuscript would be strengthened if the authors could address the points above.

We thank the reviewers for their careful review of our work. Please find below our point-by-point response to each of their comments. We made substantial changes to the manuscript based on their recommendations and think that it now provides an even more compelling set of results to support our conclusions in support of the critical role for FOXA1 plexus CRE in prostate oncogenesis, identifying in the process as new target for therapy development.

Comments and Response to reviewers in full:

Reviewer #1 (Remarks to the Author):

Functional validation of genomic elements using CRISPR/Cas9 is a major task of current functional genomics studies. FOXA1 has been found to be essential for the growth of prostate cancer cells or tumorigenesis. This study intended to validate the functions of 6 cis-regulatory elements for FOXA1 expression in prostate cancer cells and the functional genetic variants in these cis-regulatory elements. There are several concerns of this study.

1. CRISPR/Cas9 was used to examine the functions of cis-regulatory elements. However, key descriptions of this method are missing, such as sequences and locations of crRNA in each cis-regulatory element; T7 endonuclease assays were used to examine the efficiency of CRISPR/Cas9, but this assay only detects heterozygous mutations, did authors examine homozygous mutations and how? Where are the primer sequences and locations? Clearly, CRISPR/Cas9-treated cells are mixed cell populations of mutations; should publish at the level of Nature Communications, single colony with homozygous mutations would be essential for this type of functional genetic assays.

- The supplementary tables containing the sgRNA, primers and other requested information to enable reproducibility of our approach are now provided in the revised submission.
- We attempted to generate single cell clones that harbour homozygous deletions of each CRE. Despite our best efforts, our multiple attempts failed. Since deletion of each CRE results in significant reduction in cell proliferation (Figure 5d), we reason homozygous deletion are nonviable.
- As an alternative to the single colony with homozygous mutations, we derived stable clonal LNCaP (n = 5 clones) and 22Rv1 (n = 4 clones) cells that express wild-type Cas9 (at different levels; Figure 4a, b). We further supplemented this approach by generating clonal LNCaP (n = 4 clones) and 22Rv1 (n = 4 clones) cells stably expressing dCas9-KRAB (Figure 4e, f). With these clones isolated from a population

of cells, our observations still maintain the same conclusion as our initial submission, that the CREs we identified functionally regulate *FOXA1* mRNA expression. The added results upon revision are presented in Figure 4.

2. This manuscript is lack of proper citations of previous studies in the field; e.g., siRNA knockdown of FOXA1 suppressed the growth of prostate cancer cells (Fig. 2D-F) was reported in several studies previously; cis-regulatory elements of FOXA1 have been investigated in several studies previously and feedforward regulation of FOXA1 was also discovered previously.

- We added a number of citations of past work looking at the various modes of FOXA1 function and regulation. We also cited the recently published studies that demonstrate the oncogenic potential of *FOXA1* in driving prostate cancer progression to further highlight the importance of our work including: Adams *et al.*, (PMID: 31243370; 2019)¹, Parolia *et al.*, (PMID: 31243372; 2019)², Gao *et al.*, (PMID: 31324883; 2019)³, Zhang *et al.*, (PMID 27034986; 2016)⁴, Augello *et al.*, (PMID: 21934649; 2011)⁵, Yang *et al.*, (PMID: 27257062; 2016)⁶, Jin *et al.*, (PMID: 23539448; 2013)⁷, Ni *et al.*, (PMID: 23530127; 2013)⁸, Sasse *et al.*, (PMID: 25168919; 2015)⁹. These important discoveries are cited in the second and third paragraphs of the Introduction (lines 74-95) and third and fourth paragraphs of the Discussion (lines 343-355) in the manuscript.

3. It is over-interpreted for the studies of functional genetic variants in the cis-regulatory elements. It is a correlation but not a functional study. CRISPR/Cas9 would help this part.

- We attempted to edit in the genetic variants using CRISPR/Cas9 technology, but failed after multiple attempts due to the technical difficulties involved. It is worth mentioning that we are not aware of recent publications that knocked in SNVs except for the recent Adams *et al.*, (PMID: 31243370; 2019)¹ paper looking at coding *FOXA1* SNVs (co-published in *Nature* with Parolia *et al.*, (PMID: 31243372; 2019)². This is especially true in the noncoding space as key publications from other groups in the field such as Horn *et al.*, (PMID:23348503; *Science* 2013)¹⁰, Huang *et al.*, (PMID:23348506; *Science* 2013)¹¹, Fuxman Bass *et al.*, (PMID:25910213; *Cell* 2015)¹², Rheinbay *et al.*, (PMID: 28658208; *Nature* 2017)¹³, and Feigin *et al.*, (PMID: 28481342; *Nature Genetics* 2017)¹⁴ conducted similar assays as we did to study the potential impact of noncoding SNVs. Specifically, these groups tested change in transactivation potential caused by a SNV through luciferase reporter assays.
- To alternatively address the reviewer's comment, we conducted allele-specific ChIP-qPCR, a method we and others previously used to demonstrate the functional impact that noncoding SNVs on gene expression¹⁵⁻¹⁹. We find that the SNVs changing the transactivation potential of the *FOXA1* plexus CREs, namely chr14: 37887005 (A>G), chr14: 37904343 (A>T), chr14:37906009 (T>C), chr14: 37905854 (A>G) and chr14:38055269 (C>G) do impact the binding of certain transcription factors. These transcription factors include FOXA1, AR, GATA2, and FOXP1, which are known to

bind at these CREs (Figure 2b). These results are presented in Figure 6l-q.

- As an additional approach, we considered testing for allelic-imbalance in expression in tumors heterozygous for the SNVs in the *FOXAI* plexus CREs. This is a method we previously employed¹⁵ and applied to the single patient from our cohort with a mutation in a CRE predicted to regulate *FOXAI* expression. Assessing allelic imbalance in expression within this one patient (CPCG0342; n=1) harboring the chr14:37906009 T>C SNV, did not reveal any statistically significant difference between either alleles in mRNA levels for *FOXAI* versus the ratio between the mutant and wild-type allele of the SNV in the CRE. Specifically we tested 12 Single Nucleotide Polymorphisms mapping to the expressed exons of *MIPOL1*, *FOXAI* and *TTC6* and observed no significant differences between DNA and RNA allele frequencies (Revision Response Fig. 1). We will note that the variant allele frequency for the chr14:37906009 T>C (53>13 reads) SNV is 0.2, suggesting the likely maximum tumour cellularity of the sample used for RNA-seq is 40%. Hence, the dominant expression for the tested genes is likely derived from normal stromal cells, hindering our ability to call allelic-imbalance in expression from RNA-seq in this primary prostate tumour sample.

Coordinate	SNP germline	Reference Al	Variant Allele	Variant Allele Frequency	Gene	RNA-seq Var/Ref Reads Ratio
chr14:38017264	rs75073310	T	G		0.5 MIPOL1	0.48
chr14:38018435	rs1158700	T	A		0.5 MIPOL1	0.38
chr14:38018624	rs1158701	C	T		0.63 MIPOL1	0.36
chr14:38019108	rs1956439	C	T		0.52 MIPOL1	0.35
chr14:38019142	rs1956440	G	T		0.55 MIPOL1	0.39
chr14:38019334	rs117308551	A	G		0.52 MIPOL1	0.64
chr14:38019733	rs1950824	C	T		0.58 MIPOL1	0.37
chr14:38060646	rs33984772	C	T		0.61 FOXA1	0.46
chr14:38061547	rs112819884	G	C		0.45 FOXA1	0.43
chr14:38061742	rs7144658	C	T		0.59 FOXA1	0.57
chr14:38064215	rs79519281	G	C		0.58 FOXA1	0.56
chr14:38064738	rs10400689	G	T		0.44 TTC6	0.55
Coordinate	SNV somatic					
chr14:37906009	N/A	T	C	0.2		

Revision Response Figure 1 - Allelic imbalance analysis to assess the impact of a noncoding SNV mapping to *FOXAI* plexus CRE.

- In consideration of the reviewer’s comment, we modified the text to reflect the nature of our results that does not conclusively provide a direct link between the SNVs and *FOXAI* mRNA levels of the patient. These modifications can be seen throughout the manuscript such as in the results under the “Somatic mutations are capable of altering transcription factors in prostate cancer” section (lines 318-320) and the Discussion (lines 349-350).

4. It is quite confusing for the relation between Fig. 1 and the rest of studies.

- Figure 1 is meant to present the latest assessment of *FOXAI* expression and requirement in prostate cancer. This builds a rationale, in combination with the

existing literature, to deeply investigate its mode of regulation.

- To further address the reviewer's comment, we revised the manuscript surrounding Figure 1 to better connect it to the following figures under the "*FOXA1* is highly expressed and essential for prostate cancer proliferation" section.

Reviewer #2 (Remarks to the Author):

In this manuscript, Zhou and colleagues identify putative cis-regulatory elements of FOXA1 and experimentally validated 6 of them that drive FOXA1 expression in prostate cancer (PCa) cells. Additionally, they report the presence of somatic single-nucleotide mutations at these elements in PCa patients, which functionally enhance their transactivation potential in reporter assays. Although these findings are intriguing, the genomic and functional data presented in this study is very preliminary and lacks important details and validations. The conclusion that deletion of the CREs decreases FOXA1 expression and cell proliferation is appropriate, but the experimental evidence supporting this conclusion can be improved upon. In addition, the experiments used to support the conclusion that the SNVs found in patients are activating mutations at CREs of FOXA1 are suboptimal. Thus, this study requires more functional experiments (with important controls) carried out in multiple PCa cell lines. For example, functional validation of the SNVs in vitro and in vivo would provide direct relevance of those mutations to cancer progression. Specific comments/questions are listed below:

Major:

1) In Fig2B, the authors predict several of the PCa transcription factors to bind to the putative cis-regulatory elements of FOXA1. However, they do not interrogate as to which one of these factors are required for their activation. This is an important question that remains unanswered in this study.

- In Figure 2b is based on ChIP-seq data generated in prostate tumors and prostate cancer cell lines. Hence, the results are not meant to be predictions but rather actual reported binding sites for the specified transcription factors. We provide a more detailed genome browser view of the binding sites for each of the transcription factors detected at any one of the 6 CREs under investigation in Supplementary Figure 4a-f.
- To address this specifics of this comment, we considered siRNA-mediated knockdown of transcription factors known to be bound at the CREs followed by measuring *FOXA1* mRNA levels. However, we did not pursue this avenue as even reduction in *FOXA1* mRNA levels upon knocking down each factor would not provide a clear direct link to the "requirement" for any of the transcription factors toward *FOXA1* expression.
- Instead, we decided to assess if mutations found in the 6 CREs under investigation were changing the binding intensity of specific transcription factors at each CRE. This assessment was performed through allele-specific ChIP-qPCR upon transfection of wild-type and mutant CRE plasmids independently for each of the 6 CREs under

investigation. The data presented in Figure 6l-q reveals that the binding of some but not all transcription factors, including FOXA1, AR, GATA2, FOXP1 is affected by SNVs at different CREs. Since these mutations can change the transactivation of CREs (Figure 6b-k) our results collectively argue that FOXA1, AR, GATA2 and FOXP1 are strong regulatory candidates of *FOXA1* expression.

2) In fig. 4G-I, along with supp. fig. 5, the authors performed deletions of various CREs of FOXA1 using CRISPR/Cas9. However, it looks as though the efficiency of the deletions is highly variable (supp. Fig. 5a vs 5c). Thus, it is hard to tell how well the decrease in FOXA1 expression correlates with the CRE deletions as opposed to random other mutations generated by one or both sgRNAs targeting the flanking regions of the CREs. It would be stronger evidence that the deletion of those CREs decreases FOXA1 expression if the authors were able to estimate the deletion efficiency and correlate that with the reduction of expression. In addition, isolating monoclonal cells with successful deletion of the CREs and assessing the FOXA1 expression in those cells compared to wild type cells can give more solid evidence for this relationship. This comment can be applied to the combination deletion experiment as well as the proliferation assays (4G-I).

- In relation to Reviewer 1's comment about deriving single clones that have deletion of the CREs, we did attempt to do this on multiple occasions and failed due to technical challenges. We reason that it is due to the dependency of prostate cancer cells have on *FOXA1* for growth.
- Instead to probe the relationship between genome editing and reductions in *FOXA1* mRNA levels, we derived single clones from LNCaP (n=5 clones) and 22Rv1 (n=4 clones) prostate cancer cells that stably express wild-type Cas9 enzyme at different levels (Figure 4a, b).
- ImageJ densitometry was used to quantify the on-target genome editing efficiency across the different Cas9 clones via agarose gel following the T7 endonuclease I assay. We used the densitometry values from the agarose gels as proxy for on-target genome editing efficiency, and correlated these values with *FOXA1* mRNA levels measured from the same cells upon deletion. We observed an overall significant Pearson's correlation of 0.49. When we calculated the correlation based on each CRE, there is an even greater correlation between the genome editing efficiency and *FOXA1* mRNA levels. The data is presented in Supplementary Figure 5a, b (lines 202-206).
- The take-home message upon adding results from different Cas9-based approaches remains that we identified a set of six CREs that regulate *FOXA1* mRNA expression.

3) The authors use a pair of sgRNAs to disrupt the CREs. This approach is very inefficient as it requires both the sgRNAs to be present inside the same cell for the deletion to occur, thus ensuing in only a minute fraction of the cells harboring a successful deletion. Additionally, many of the cells would simply harbor in-dels at the cut sites rather than deletions of the intermediate DNA (which is also validated by the T7 endonuclease assays - a deletion would not generate a mismatch). This is particularly true when sequentially targeting multiple CREs in the cells. Thus, the authors should corroborate the CRE-disruption data using orthogonal

approaches, such as the dCas9-KRAB system.

- We now include results from both LNCaP and 22Rv1 cell lines stably expressing dCas9-KRAB. Through transient transfection of the sgRNAs that target each CRE under investigation into these stable clonal lines, we observed a significant decrease in *FOXAI* mRNA expression (Figure 4e-h). These new results are also presented in the results section of the manuscript under the “Clonal dCas9-KRAB disruption of *cis*-regulatory elements resulted in reduced *FOXAI* mRNA expression” subheading (lines 207-217).

4) Related to above, what is the effect of disrupting these CREs on other TAD genes, such as *TTC6*?

- We now included mRNA expression of *MIPOLI* and *TTC6* upon deletion of the CREs. Deletion of CRE1 and CRE2 resulted in the significant reduction in *MIPOLI* mRNA expression (Supplementary Figure 10A). Deletion of CRE1-6 all significantly reduced *TTC6* mRNA expression (Supplementary Figure 10B). The latter is expected because *TTC6* shares its promoter with *FOXAI*, being transcribed from the opposite strand. These results are also now part of the manuscript (lines 218-228).

5) Additionally, are there other SNVs within this *FOXAI* TAD genomic region? The author should report all the SNVs found in these region and test that non-enhancer SNVs are not functionally relevant in their reported assays.

- There are a total of 75 SNVs that are within this TAD (64 SNVs directly within the TAD, and the other 11 SNVs accounting for the 40 kb resolution of the TAD call), which we now report in Supplementary Table 1.
- We used the dual luciferase reporter assay relying on plasmids to determine the ability of SNVs to drive changes in the transactivation potential. Despite luciferase reporter assay being a robust method to interrogate the impact of a single nucleotide change via a luminescence readout, the plasmids are a piece of naked DNA, as opposed to a chromatinized template. So while we can test any non-CRE SNV, this will be in a n irrelevant chromatin context. This is important, because if we look at Figure 2b, we see that the only SNVs in accessible chromatin are the SNVs we tested through luciferase reporter assays (i.e other SNVs are in non-accessible chromatin). We instead adjusted the text of the manuscript to clearly state that we detected changes in the transactivation potential of only 6 out of the 10 SNVs tested that map to open chromatin. Hence, the presence of an SNV in active chromatin is not sufficient to infer its function. This is an important statement that we feel more directly assesses the role of SNVs in prostate tumours.

6) It is possible that any genomic cut within the *FOXAI* TAD could ensue in modest down-regulation of *FOXAI*. Thus, the authors should include an important control sgRNA pair that cuts within the TAD but not at a putative CRE element.

- To address this comment, we designed *three* pairs of sgRNAs that target three different CREs within the TAD (i.e termed Within TAD #1, #2 and #3) that are not predicted to regulate *FOXA1* expression. Moreover, we tested these controls through *three* complementary strategies:

- 1) Transient transfection-based ribonucleoprotein complex CRISPR/Cas9,
- 2) Lentiviral-based clones stably expressing wild-type CRISPR/Cas9, and
- 3) Lentiviral-based clones stably expressing dCas9-KRAB.

- Collectively, we demonstrate that the deletion of these three additional CREs not predicted to interact with the *FOXA1* promoter did not impact *FOXA1* expression. Results are presented in Figure 4c, d, g, h and Figure 5a.

7) For the sequential disruption of CREs, were the negative control cuts also compounded? These targeted cutting within the FOXA1 TAD could simply result in collapsing with 3D structure of the TAD instead of the intended focused deletion of the CRE element alone. Thus, it is important to use intra-TAD negative control sgRNA pairs and sequentially compound them when deleting multiple CREs together.

- Our initial submission used a single negative control cut. In line with the reviewer's comment, we repeated the experiment using combinations of negative controls using sgRNA targeting two separate intra-TAD regions used to address comment 2.6. Specifically, we measured *FOXA1* mRNA levels upon the double deletions of:

- CRE1 + CRE2,
- CRE2 + CRE4,
- CRE4 + CRE1,
- IntraTAD Negative #1 + IntraTAD Negative #2,
- IntraTAD Negative #2 + IntraTAD Negative #3,
- IntraTAD Negative #3 + IntraTAD Negative #1.

- Our results presented in Figure 5a-b do show a minor reduction of *FOXA1* expression following the double deletion of negative regions compared to the single deletions. In contrast however, when we conducted double deletions of CRE1, 2 and 4 in combinations, there is a further significant reduction in *FOXA1* mRNA levels when compared to single deletions (Supplementary Figure 10). As such, our observations based on our revised experiments still holds that 1) the CREs work collaboratively to maintain *FOXA1* mRNA levels and 2) combinatorial deletions significantly further reduces *FOXA1* mRNA levels compared to single CRE deletions. These results are also presented in the manuscript (lines 254-262).

8) Since CREs affect the expression of the target gene through chromatin looping, plasmid-based firefly luciferase assays are not the best approach to testing this phenomenon, due to the extragenomic nature of plasmids.

- The reviewer is correct to point out that plasmid-based assays do not fully recapitulate the chromatin context relevant to gene expression. Instead they allow to study the specific transactivation potential of CREs. We have revised the manuscript to clearly indicate that the luciferase assays were employed to assess if SNVs could alter the transactivation potential of CREs selected based on their predicted contact with *FOXA1* expression (Figure 3) and their ability to regulate *FOXA1* expression drawn from their knockout using CRISPR/Cas9 and their knock-down using CRISPR/dCas9-KRAB (Figure 4).

9) Most importantly, almost every experiment in this study is only carried out in the LNCaP cells. The authors need to use other AR/FOXA1+ PCa models.

- To address this comment, we conducted and replicated experiments in 22Rv1 cells, another AR+ and FOXA1+ prostate cancer cell line. We derived and conducted CRISPR/Cas9 experiments in both LNCaP (n=5 clones) and 22Rv1 (n=4 clones) cells that stably expressing wild-type Cas9 protein (results shown in Figure 4a-d). We also derived and conducted dCas9-KRAB experiments in both LNCaP (n=4) and 22Rv1 clones (n=4) clones that stably express dCas9-KRAB fusion protein (results shown in figure 4e-h). Collectively, results from the 22Rv1 cell line are consistent with those observed from LNCaP cell lines demonstrating that our observations are not simply artefacts of a single model system.

10) The authors report the presence of SNVs within these cis-elements in patient tumors; however, the functional validation of the SNVs found in patients is weak. Are these event hetero or homozygous? Are they within known motifs of the TFs there were predicted to bind to these elements in the in-silico analyses?

- The SNVs found within the tumours are heterozygous as they were called based on germline genomes derived from blood, as previously reported in Fraser *et al.* (PMID 28068672; *Nature* 2017)²⁰. Using Find Individual Motif Occurrences (FIMO)²¹ searching for transcription factor binding motifs from the HOCOMOCO database (v11)²². Briefly, we conducted FIMO analyses using a 31 bp window centered on the SNV (position 16 of 31 nucleotides). We indeed do see the SNVs we interrogated in our study map to motifs of transcription factors that are known to be bound at these CREs. For example, we observe the chr14:37905854 (A>G), chr14:37906009 (T>C) and chr14:38055908 (T>C) SNVs directly map to a Forkhead binding motif (Revision Response Fig. 2). The Forkhead binding motif is known to be recognized and bound by transcription factors of the Forkhead family such as FOXA1 and FOXP1 (Figure 2b). As another example, we observe the chr37904343 (A>T) SNV mapping directly to the Homeobox motif recognized by HOXB13. These findings further support our allele-specific ChIP-qPCR data that demonstrates the SNVs can modulate the binding of transcription factors compared to the wild-type sequence (Figure 6l-q).

motif_id	motif_alt_id	mutation	start	stop	strand	score	p-value	q-value	matched_sequence
FOXA2_HUMAN.H11MO.0.A		chr14: 37905854	7	18	-	11.2576	7.35e-05	0.00147	TATTTACTAAGC
FOXA3_HUMAN.H11MO.0.B		chr14: 37905854	8	20	-	10.4394	0.000108	0.00195	AATATTTACTAAG
FOXA1_HUMAN.H11MO.0.A		chr14: 37905854	7	18	-	10.2537	0.000134	0.00269	TATTTACTAAGC
CRX_HUMAN.H11MO.0.B		chr14: 37905854	1	13	+	9.36364	0.000231	0.00416	taaaaagcttagt
FOXM1_HUMAN.H11MO.0.A		chr14: 37905854	7	18	-	9.08955	0.000269	0.00538	TATTTACTAAGC
MAFK_HUMAN.H11MO.0.A		chr14: 37905854	1	18	+	2.51515	0.000735	0.00588	taaaaagcttagtaaata
MAFG_HUMAN.H11MO.0.A		chr14: 37905854	1	18	+	-1.5	0.000766	0.00613	taaaaagcttagtaaata
OTX2_HUMAN.H11MO.0.A		chr14: 37905854	3	13	+	6.87879	0.000932	0.0205	aaaagcttagt
FOXC1_HUMAN.H11MO.0.C		chr14: 37906009	3	17	+	10.0787	5.42e-05	0.00184	gaatgtttacctata
FOXA3_HUMAN.H11MO.0.B		chr14: 37906009	4	16	+	11.3182	6.41e-05	0.00243	aatgtttacctat
FOXA2_HUMAN.H11MO.0.A		chr14: 37906009	6	17	+	11.3788	6.71e-05	0.00269	tgtttacctata
FOXJ2_HUMAN.H11MO.0.C		chr14: 37906009	6	15	+	11.7826	6.81e-05	0.00299	tgtttaccta
FOXA1_HUMAN.H11MO.0.A		chr14: 37906009	6	17	+	11.3433	7.22e-05	0.00289	tgtttacctata
FOXK1_HUMAN.H11MO.0.A		chr14: 37906009	6	15	+	10.72	7.94e-05	0.00349	tgtttaccta
FOXM1_HUMAN.H11MO.0.A		chr14: 37906009	6	17	+	10.5522	9.19e-05	0.00368	tgtttacctata
ZFP42_HUMAN.H11MO.0.A		chr14: 37904343	15	26	-	11.803	3.91e-05	0.00157	AACATCCATTTT
HXB13_HUMAN.H11MO.0.A		chr14: 37904343	9	18	-	11.1343	7.27e-05	0.0032	TTTTATAAGT
NR1I3_HUMAN.H11MO.0.C		chr14: 38056977	1	18	-	9.9	8.18e-05	0.00229	GGAGTGAGCATAGGACAG
SNAI1_HUMAN.H11MO.0.C		chr14: 38055269	5	12	-	14.374	9.9e-06	0.000475	CCAGGTGG
ZIM3_HUMAN.H11MO.0.C		chr14: 38055269	8	26	+	7.69697	9.8e-05	0.00255	CCTGGATTCTGATGTATTC
AIRE_HUMAN.H11MO.0.C		chr14: 38127842	6	23	+	12.0899	2.87e-05	0.000636	TTGGGTTATTTGGGTGAA
AIRE_HUMAN.H11MO.0.C		chr14: 38127842	5	22	+	11.5618	4.54e-05	0.000636	ATTGGGTTATTTGGGTGA
MEF2D_HUMAN.H11MO.0.A		chr14: 38055908	9	20	+	12.8657	1.96e-05	0.000783	TTCTATTATAAA
MEF2A_HUMAN.H11MO.0.A		chr14: 38055908	9	21	+	12.4925	2.62e-05	0.000997	TTCTATTATAAA
MEF2C_HUMAN.H11MO.0.A		chr14: 38055908	9	21	+	12.2687	3.06e-05	0.00116	TTCTATTATAAA
FOXO1_HUMAN.H11MO.0.A		chr14: 38055908	16	27	-	11.5273	3.64e-05	0.00145	TTCTGTTTATA
FOXO3_HUMAN.H11MO.0.B		chr14: 38055908	16	25	-	11.5606	6.14e-05	0.0027	CCTGTTTATA
FOXO2_HUMAN.H11MO.0.C		chr14: 38055908	16	24	-	11.3182	7.17e-05	0.0033	CTGTTTATA
ETV4_HUMAN.H11MO.0.B		chr14: 38055908	20	30	+	11.3182	7.28e-05	0.00306	AACAGGAAGAT
MEF2B_HUMAN.H11MO.0.A		chr14: 38055908	7	20	+	10.6418	8.57e-05	0.00309	AATTCTATTATAA
NKX31_HUMAN.H11MO.0.C		chr14: 38036543	11	22	-	10.7248	8.63e-05	0.00345	CTAAAGTACTTT
GATA3_HUMAN.H11MO.0.A		chr14: 37887437	3	13	+	12.6364	1.6e-05	0.000674	AGAGATAAAA

Revision Response Figure 2 - SNVs at CREs map to binding motifs known to be bound by transcription factors that bind at the CREs. Individual motif occurrences were searched within the 31 bp sequence centered on the SNV coordinate (flanked by 15 bp each side) against the HOCOMOCO database of 402 motifs (v11) ²².

11) What is the expression of FOXA1 in these patients relative to WT enhancer controls. Even though the incorporation of these SNVs seemed to increase expression, it may not be a result of chromatin looping. A better way to test this hypothesis is to use CRISPR/Cas9 with guide RNAs targeting the CREs in a cell model, isolate monoclonal cells with the mutations seen in patients, and observe the expression of FOXA1.

- As suggested by the reviewer, an alternative is to use the CRISPR/Cas9 technology to create a heterozygous model for the SNVs under study in prostate cancer cell lines. Despite our best efforts, we were unable to achieve successful knock-in of SNVs. During the revision period alone, we plated 3-4 96-well plates per SNV upon attempted knock-in (i.e chr14:37906009 T>C; chr14: 37905854 A>G; chr14: 38036543 A>G) at the bulk level upon serial dilution, aiming for 1 cell per well. We prioritized on these three SNVs based on 1) significant change in transactivation potential and 2) its distance to a Cas9 cut site, which has been shown to be important for knock-in efficiency (Integrated DNA Technologies: “Homology-directed repair using Alt-R CRISPR-Cas9 system and Ultramer Oligos” protocol). This resulted in about 50-90 clones that grew out from the 96-well plates, dependent on the SNV, for

each SNV. Despite our efforts, we did not successfully isolate any clones that harbour the SNVs.

- In relation to the comment and our response to Reviewer 1 (comment 1.3), we are not aware of recent publications that knocked in SNVs in the noncoding space. In addition to our initial submission, we did conduct allele-specific ChIP-qPCR to assess whether the SNVs modulate transcription factor binding. We next also probed for allelic imbalance with the CPG0342 patient that has RNA-seq available, but observed no significant differences in allele-biased *FOXA1* expression, likely due to the limited cellularity of the sample (see response to comment 1.3). Taken together, we tried to address the reviewer's suggestion, but the current state of CRISPR/Cas9 technology and technical limitations were too great for us to isolate any clones for this revision and this question remains unanswered.

Reviewer #3 (Remarks to the Author):

In the present study, Zhou S. et al. identify 6 putative FOXA1 cis-regulatory elements using single nucleotide variant calls from whole genome sequencing data from primary prostate cancer samples, paired with publicly available DHS, Hi-C, and TADs datasets of the prostate cancer cell line. They further singly or sequentially delete these elements in LNCaP cells with CRISPR/Cas9 approaches, showing that these deletions result in the reduction of FOXA1 expression and cell growth. They also study the function of 10 single-nucleotide variants in these elements using luciferase reporter assays and find that 6 of 10 SNVs have gain-of-function activity.

Overall, this study addresses the regulation of FOXA1, a critical transcription factor, in prostate cancer, and includes multiple novel findings, including the first delineation of the complex network of cis-regulatory elements of FOXA1. The manuscript is well written and well presented. At the same time, there are several issues that could be addressed to improve the manuscript. These are described below:

1) The authors screened for somatic SNVs from whole genome sequences of 200 primary prostate cancers, thus identifying 10 total SNVs in FOXA1 CREs. Given that the frequency of genomic alterations is significantly higher in metastatic prostate cancer compared to primary disease, could the authors provide a genomic analysis using whole genome sequencing data from metastatic castration-resistant prostate cancer (mCRPC)? Recently, there have been several different whole genome sequencing studies published in mCRPC, including several in Cell, and a study from the Netherlands recently reported on BioRxiv. I realize that it would be beyond the scope of this project to functionally characterize additional SNVs found with this analysis, but it would be good to understand how many more FOXA1s would be nominated based on SNV calls from metastatic disease.

- We agree with the value of expanding our analysis to mutations called in metastatic castration-resistant prostate cancer. However, our approach relies on using matched

annotation of the cancer genome, specifically mutations and cis-regulatory elements. While the manuscripts mentioned by the reviewer, including Quigley *et al.* (PMID:30033370; *Cell* 2018)²³ and the van Dessel *et al.* manuscript on bioRxiv²⁴, provide details on the genetic alterations in mCRPC, there are no maps of cis-regulatory elements in mCRPC. We consider this to be a limitation to properly address the reviewer's comment, considering that we and others previously demonstrated that *cis*-regulatory elements differ between primary and drug resistant cancers²⁵. If the CREs we identified indeed regulate *FOXA1* in the mCRPC setting, then it would be intriguing to know whether noncoding SNVs acquired in mCRPC do also map to the CREs. This is an intriguing hypothesis especially now we know from the recent publications that the CRE2, CRE3, CRE4 and CRE5 we describe here in regulation of *FOXA1* are targets of various structural variation (i.e amplification, duplication and translocation) in prostate cancer^{1,2,23}. As such we agree with the reviewer and appreciate the suggestion, and we think this would be an exciting question to ask in a distinct manuscript.

2) The authors show that the nominated SNVs affect the transactivation potential of CREs via luciferase assays. Do any of these SNVs have a large enough effect such that there is an impact on cell growth/proliferation? (i.e., the one or two with the highest reporter activity?)

- To address this comment, we tried to knock in the SNVs (i.e chr14:37906009 T>C; chr14: 37905854 A>G; chr14: 38036543 A>G) that 1) have significant luciferase reporter activity differences relative to wild-type, and 2) have the highest probability of homologous recombination based on its presence near the cut site of the Cas9 enzyme as discussed in the literature and manufacturer's instructions (Integrated DNA Technologies: "Homology-directed repair using Alt-R CRISPR-Cas9 system and Ultramer Oligos" protocol). Upon screening 50-90 clones that grew out for each SNV (depending on the SNV), we were not able to detect any successful editing. Despite our best efforts, we could not overcome the technical limitations of CRISPR/Cas9 technology to address this suggestion. It is worth mentioning that we are also not aware of other publications that consistently knocked in SNVs in the noncoding space.
- In relation to the other two reviewers' comments (i.e 1.3 and 2.11), we could not provide a conclusive claim based on our efforts about the phenotypic impact of these SNVs on target gene expression or impact on growth. We have since revised the manuscript to reflect our efforts to provide the strongest possible mechanistic link between the noncoding SNVs mapping to the CREs and *FOXA1* mRNA levels.

3) As a complementary strategy to assess gain of function of *FOXA1* based on the nominated CREs (or the SNVs with the CREs), have the authors check the downstream genes regulated by *FOXA1*?

- We proceeded to harvest mRNA from the five LNCaP Cas9 clones upon deletion of CREs. We then interrogated the mRNA expression of highly expressed target genes

downstream of *FOXA1* signalling through reverse transcriptase qPCR, namely *GRIN3A*, *ACPP* and *SNAI2* previously reported to be regulated by FOXA1. Results presented in Supplementary Figure 7 are summarized below.

- *GRIN3A* mRNA expression:
 - Δ CRE1 ~120.6%
 - Δ CRE2 ~157%
 - Δ CRE3 ~140.9%
 - Δ CRE4 ~131.6%
 - Δ CRE5 ~121.2%,
 - Δ CRE6 ~122.9%,
 - Δ *FOXA1* ~217.8%
 - Δ AAVS1 ~106.2%
 - Δ Chr14- ~106.4%
- *ACPP* mRNA expression:
 - Δ CRE1 ~73.5%
 - Δ CRE2 ~62.5%
 - Δ CRE3 ~69.6%
 - Δ CRE4 ~75.6%
 - Δ CRE5 ~70.9%,
 - Δ CRE6 ~74.6%,
 - Δ *FOXA1* ~52.2%
 - Δ AAVS1 ~96.5%
 - Δ Chr14- ~96.4%
- *SNAI2* mRNA expression:
 - Δ CRE1 ~190%
 - Δ CRE2 ~162.8%
 - Δ CRE3 ~147.5%
 - Δ CRE4 ~133.3%
 - Δ CRE5 ~137.3%,
 - Δ CRE6 ~120.8%,
 - Δ *FOXA1* ~266.7%
 - Δ AAVS1 ~100.2%
 - Δ Chr14- ~96.2%

Hence, deletion of CREs part of the *FOXA1* plexus also impact the expression of FOXA1-regulated genes found on other chromosomes (*GRIN3A* on chromosome 9, *ACPP* on chromosome 3, *SNAI2* on chromosome 8). We also observed that the disruption of specific CREs (e.g CRE6) consistently have a weaker impact than other CREs (e.g CRE2) on both *FOXA1* mRNA levels and the mRNA levels of these downstream FOXA1-regulated genes. These observations can be made from looking at Figure 4c, d, g, h, 5a and Supplementary Figure 10d, e, f. These results are also described in the manuscript (lines 229-240).

4) As the authors mentioned, FOXA1 promotes prostate cancer progression partially through AR signaling. Have the authors check the AR signaling when disrupting their nominated

CREs?

- To address the reviewer's comment, we used the mRNA harvested upon deletion of the CREs to check the mRNA expression of *AR*. We observed a trend but not significant changes in *AR* expression upon deleting the promoter of *FOXA1* (Rebuttal Figure 3).

Revision Response Figure 3: *AR* mRNA levels normalized to housekeeping *TBP* mRNA levels upon CRISPR/Cas9-mediated deletion of each CRE using each LNCaP clone stably expressing Cas9 protein (n=3 clones). None of them are significantly different compared to AAVS1, a negative control region.

5) The putative FOXA1 CRE1 and CRE2 are located in the gene *MIPOL1*. Have the authors check the impact on *MIPOL1* when using CRISPRn to delete CRE1/2?

- As stated by Reviewer 2 in comment 2.4, *MIPOL1* is one of the genes within the same TAD harbouring *FOXA1*, and indeed CRE1 and CRE2 are located within the *MIPOL1* gene. To address the current comment, we conducted reverse transcriptase qPCR using RNA harvested upon deletion of CRE1 and CRE2 in LNCaP prostate cancer cells (n = 5 clonal lines) that stably express wild-type Cas9. Specifically, we observe an average of 38.4% and 48.4% significant reduction in *MIPOL1* mRNA levels upon deletion of CRE1 and CRE2, respectively. In contrast, the deletion of CRE3-6 had no significant impact on *MIPOL1* mRNA levels. The results are now presented in Supplementary Figure 10a. These results are described in the manuscript (lines 218-228).

Minor issues include the following:

6) In Figure 1, what are the FOXA1 mRNA levels with the siRNA knockdown?

- Across the three biological replicates, the *FOXA1* mRNA expression was knocked down by 89% and 89% by siRNA #1 and siRNA #2, respectively. The data is now presented in Supplementary Figure 3b.

7) The authors use CRISPRn to delete CREs. Have the authors quantified the deletion efficiency beside the T7 assay shown in Supplemental Figure 5? Will the different efficiencies impact the FOXA1 expression levels and cell growth in Figure 4?

- In relation to Reviewer 2's comment (2.2), we addressed this concern by using genomic DNA and RNA from stable LNCaP wild-type Cas9 clonal cell lines and interrogating the correlation between the on-target genome editing efficiency and reduction in *FOXA1* mRNA expression. We see that there is a significant Pearson's correlation of 0.49 across the deletion of all CREs and resultant *FOXA1* mRNA expression (Supplementary Fig. 5a). When we further calculated correlation based a per CRE level, we see an even stronger Pearson's correlation (Supplementary Fig. 5b). It is also important to note that on top of the correlation, the genome editing efficiency is consistently high, suggesting there may be an absolute contribution of each CRE to *FOXA1* mRNA expression regulation.

8) Perhaps I missed these, but I think the supplementary tables are missing.

- The supplementary tables as indicated in the main text should now be attached to this revised submission.

Overall, this is a provocative study with novel findings; the manuscript would be strengthened if the authors could address the points above.

- We thank the reviewer for the positive and enthusiastic comments.

References:

1. Adams, E. J. *et al.* FOXA1 mutations alter pioneering activity, differentiation and prostate cancer phenotypes. *Nature* **571**, 408–412 (2019).
2. Parolia, A. *et al.* Distinct structural classes of activating FOXA1 alterations in advanced prostate cancer. *Nature* **571**, 413–418 (2019).
3. Gao, S. *et al.* Forkhead domain mutations in FOXA1 drive prostate cancer progression.

Cell Res. (2019). doi:10.1038/s41422-019-0203-2

4. Zhang, G. *et al.* FOXA1 defines cancer cell specificity. *Science Advances* **2**, e1501473 (2016).
5. Augello, M. A., Hickey, T. E. & Knudsen, K. E. FOXA1: master of steroid receptor function in cancer. *EMBO J.* **30**, 3885–3894 (2011).
6. Yang, Y. A. *et al.* FOXA1 potentiates lineage-specific enhancer activation through modulating TET1 expression and function. *Nucleic Acids Res.* **44**, 8153–8164 (2016).
7. Jin, H.-J., Zhao, J. C., Ogden, I., Bergan, R. C. & Yu, J. Androgen receptor-independent function of FoxA1 in prostate cancer metastasis. *Cancer Res.* **73**, 3725–3736 (2013).
8. Ni, M. *et al.* Amplitude modulation of androgen signaling by c-MYC. *Genes Dev.* **27**, 734–748 (2013).
9. Sasse, S. K. & Gerber, A. N. Feed-forward transcriptional programming by nuclear receptors: regulatory principles and therapeutic implications. *Pharmacol. Ther.* **145**, 85–91 (2015).
10. Horn, S. *et al.* TERT promoter mutations in familial and sporadic melanoma. *Science* **339**, 959–961 (2013).
11. Huang, F. W. *et al.* Highly recurrent TERT promoter mutations in human melanoma. *Science* **339**, 957–959 (2013).
12. Fuxman Bass, J. I. *et al.* Human gene-centered transcription factor networks for enhancers and disease variants. *Cell* **161**, 661–673 (2015).
13. Rheinbay, E. *et al.* Recurrent and functional regulatory mutations in breast cancer. *Nature* **547**, 55–60 (2017).
14. Feigin, M. E. *et al.* Recurrent noncoding regulatory mutations in pancreatic ductal adenocarcinoma. *Nat. Genet.* **49**, 825–833 (2017).
15. Bailey, S. D. *et al.* Noncoding somatic and inherited single-nucleotide variants converge

- to promote ESR1 expression in breast cancer. *Nat. Genet.* (2016). doi:10.1038/ng.3650
16. Zhang, X., middle dot, Bailey, S. D., Moore, J. H. & Lupien, M. Integrative functional genomics identifies an enhancer looping to the SOX9 gene disrupted by the 17q24.3 prostate cancer risk locus. *Genome Research* **22**, 1437–1446 (2012).
 17. Cowper-Sal lari, R. *et al.* Breast cancer risk-associated SNPs modulate the affinity of chromatin for FOXA1 and alter gene expression. *Nat. Genet.* **44**, 1191–1198 (2012).
 18. Bailey, S. D. *et al.* ZNF143 provides sequence specificity to secure chromatin interactions at gene promoters. *Nat. Commun.* **2**, 6186 (2015).
 19. Zhou, S., Treloar, A. E. & Lupien, M. Emergence of the Noncoding Cancer Genome: A Target of Genetic and Epigenetic Alterations. *Cancer Discov.* **6**, 1215–1229 (2016).
 20. Fraser, M. *et al.* Genomic hallmarks of localized, non-indolent prostate cancer. *Nature* **541**, 359–364 (2017).
 21. Grant, C. E., Bailey, T. L. & Noble, W. S. FIMO: scanning for occurrences of a given motif. *Bioinformatics* **27**, 1017–1018 (2011).
 22. Kulakovskiy, I. V. *et al.* HOCOMOCO: expansion and enhancement of the collection of transcription factor binding sites models. *Nucleic Acids Res.* **44**, D116–25 (2016).
 23. Quigley, D. A. *et al.* Genomic Hallmarks and Structural Variation in Metastatic Prostate Cancer. *Cell* **174**, 758–769.e9 (2018).
 24. van Dessel, L. F. *et al.* The genomic landscape of metastatic castration-resistant prostate cancers reveals multiple distinct genotypes with potential clinical impact.
doi:10.1101/546051
 25. Magnani, L. *et al.* Genome-wide reprogramming of the chromatin landscape underlies endocrine therapy resistance in breast cancer. *Proc. Natl. Acad. Sci. U. S. A.* **110**, E1490–9 (2013).

REVIEWERS' COMMENTS:

Reviewer #2 (Remarks to the Author):

In the revised manuscript, the authors have addressed most of the major comments and addition of new data has strengthened their primary conclusions. Though certain ideal methodologies were not achievable, the authors have made great efforts to supplement their observations and form a cohesive and convincing study.

The authors, however, should strongly consider acknowledging and incorporating the recent discovery of a key FOXA1 enhancer (reported by Parolia et al., 2019; already cited), which was annotated as FOXMIND. Was this enhancer element functionally tested in this study? Do the two studies agree in their primary conclusion about the enhancer's transcriptional role? It is important that the authors compare their findings to the findings made by Parolia et al. and comment on how structural rearrangements within the FOXA1 locus (i.e. class3 alterations in the Parolia et. al. study) affect the plexus of FOXA1 enhancers defined in this study. Because this study is a detailed catalog of FOXA1 cis-regulatory elements in prostate cells, it would be highly relevant to discuss how this study's findings compares with or complements other, published seminal findings on the exact same topic. In addition to the rare SNVs, the highly-recurrent FOXA1 locus rearrangements could provide hints towards the influence of FOXA1 enhancers on prostate oncogenesis and could significantly improve the current manuscript.

Minor comments:

- 1) In Figure 5 (and corresponding Results section), the cell model used for these experiments was never mentioned. The figure legends did indicate that the experiments were done on LNCaP cells, but it would be nice for the model to be explicitly stated in the Results and on the Figure.
- 2) The phenotypic observations (Figures 5C-D) could become more generalizable if the authors could repeat results in other models such as VCaP.

Reviewer #3 (Remarks to the Author):

As described in this revised manuscript, Zhou et al have conducted a number of experiments to address my previous critiques. While a few of my critiques could not be addressed (due to lack of available data in publicly available datasets, etc), the authors have overall provided satisfactory responses to the majority of my earlier queries.

It is becoming clear that FOXA1 is a critical oncogene in prostate cancer, and this study includes novel findings regarding the cis-regulatory network of FOXA1. I do not have any further critiques on this manuscript, and commend the authors for their effort on this study.

Point-by-point response to Reviewers' comments:

"REVIEWERS' COMMENTS:

Reviewer #2 (Remarks to the Author):

In the revised manuscript, the authors have addressed most of the major comments and addition of new data has strengthened their primary conclusions. Though certain ideal methodologies were not achievable, the authors have made great efforts to supplement their observations and form a cohesive and convincing study.

The authors, however, should strongly consider acknowledging and incorporating the recent discovery of a key FOXA1 enhancer (reported by Parolia et al., 2019; already cited), which was annotated as FOXMIND. Was this enhancer element functionally tested in this study? Do the two studies agree in their primary conclusion about the enhancer's transcriptional role? It is important that the authors compare their findings to the findings made by Parolia et al. and comment on how structural rearrangements within the FOXA1 locus (i.e. class3 alterations in the Parolia et. al. study) affect the plexus of FOXA1 enhancers defined in this study. Because this study is a detailed catalog of FOXA1 cis-regulatory elements in prostate cells, it would be highly relevant to discuss how this study's findings compares with or complements other, published seminal findings on the exact same topic. In addition to the rare SNVs, the highly-recurrent FOXA1 locus rearrangements could provide hints towards the influence of FOXA1 enhancers on prostate oncogenesis and could significantly improve the current manuscript.

- We agree with the reviewer's comment. With recent publications and our work in combination, it is becoming increasingly clear that the noncoding space surrounding *FOXA1* contributes significantly to prostate cancer.
- To specifically address this comment, we confirm that the CRE3 we identified in our work indeed maps to the FOXMIND enhancer reported in the Parolia *et al* manuscript (PMID: 31243372; 2019). Our functional dissection indicates that upon deletion and/or repression of CRE3, *FOXA1* and *TTC6* mRNA expression levels drop significantly (Figure 4c,d,g,h, Figure 5a,b, Supplementary Figure 7b), in accordance to Parolia *et al.*'s results (their Extended Data Figure 9d-h).
- Further agreeing with Parolia *et al.*'s work on FOXMIND (their Extended Data Figure 9e), our data indicates that deletion and/or repression of CRE3 does not affect *MIPOL1* mRNA expression (Supplementary Figure 7a).
- In agreement with the growth essentiality of *FOXA1*, our results also indicate that CRE3 is important for prostate cancer cell proliferation.
- To further comment on Parolia *et al.*'s findings regarding FOXMIND, we suspect these genetic alterations (i.e duplication and translocation) targeting the noncoding space of *FOXA1* drive aberrant *cis*-regulatory activity and *FOXA1* overexpression for disease progression. In support of this hypothesis, we also discussed in the paper that CRE2 (mapping to *MIPOL1*) is a target of tandem duplication in metastatic castration resistant prostate cancer setting (reported in Quigley *et al.* Cell, 2018).
- The sum of these observations are now fully integrated within our revised manuscript in the Discussion section.

Minor comments:

1) In Figure 5 (and corresponding Results section), the cell model used for these experiments was never mentioned. The figure legends did indicate that the experiments were done on LNCaP cells, but it would be nice for the model to be explicitly stated in the Results and on the Figure.

- To address the reviewer's comment, we added further descriptions to Figure 5 in the Results section and Figure itself.

2) The phenotypic observations (Figures 5C-D) could become more generalizable if the authors could repeat results in other models such as VCaP.

- In agreement with the reviewer's suggestion, we do suspect that deletion of the CREs would impact VCaP cell proliferation, as we report in LNCaP in Figure 5c and d considering the similar responses observed across multiple assays between LNCaP, 22Rv1 and VCaP cell lines, as we present in Figures 1 and 4.

We thank the reviewer for their suggestions into improving the quality of our paper!

Reviewer #3 (Remarks to the Author):

As described in this revised manuscript, Zhou et al have conducted a number of experiments to address my previous critiques. While a few of my critiques could not be addressed (due to lack of available data in publicly available datasets, etc), the authors have overall provided satisfactory responses to the majority of my earlier queries.

It is becoming clear that FOXA1 is a critical oncogene in prostate cancer, and this study includes novel findings regarding the cis-regulatory network of FOXA1. I do not have any further critiques on this manuscript, and commend the authors for their effort on this study.

- We thank the reviewer for their efforts and suggestions to improve our work and manuscript!